# Thermalizer: Stable autoregressive neural emulation of spatiotemporal chaos

Christian Pedersen [1 2]   Laure Zanna [1 2]   Joan Bruna [1 2]

## Abstract

Autoregressive surrogate models (or *emulators*) of spatiotemporal systems provide an avenue for fast, approximate predictions, with broad applications across science and engineering. At inference time, however, these models are generally unable to provide predictions over long time rollouts due to accumulation of errors leading to diverging trajectories. In essence, emulators operate out of distribution, and controlling the online distribution quickly becomes intractable in large-scale settings. To address this fundamental issue, and focusing on time-stationary systems admitting an invariant measure, we leverage diffusion models to obtain an implicit estimator of the score of this invariant measure. We show that this model of the score function can be used to stabilize autoregressive emulator rollouts by applying on-the-fly denoising during inference, a process we call *thermalization*. Thermalizing an emulator rollout is shown to extend the time horizon of stable predictions by two orders of magnitude in complex systems exhibiting turbulent and chaotic behavior, opening up a novel application of diffusion models in the context of neural emulation.

## 1. Introduction

The modeling of dynamical systems is a cornerstone task in the physical sciences and engineering, with applications across weather (Pathak et al., 2022; Watt-Meyer et al., 2023; Lam et al., 2023; Lang et al., 2024) and climate modeling (Kochkov et al., 2024; Subel & Zanna, 2024; Lupin-Jimenez et al., 2025). The standard approach to modeling these systems is to solve the underlying partial differential equation (PDE) describing the system using numerical methods. In the case of high-dimensional systems, the computational

cost of numerical methods becomes extremely large. Turbulent fluid flows, for example, involve dynamical coupling across length scales spanning orders of magnitude, and accurately modelling the full dynamical range is often intractable for many important applications.

Deep learning (DL) methods have been applied in various ways to mitigate the cost of obtaining PDE solutions. One approach is to couple a coarse-resolution numerical solver with a DL component, either as a learned correction (Kochkov et al., 2021) or as an additional forcing term in the PDE, formulated using the large-eddy simulation framework (Duraisamy et al., 2019). DL models have also been applied as surrogate models, or *emulators*, replacing the numerical scheme entirely, and leveraging the speed of graphical processing units (GPUs) to provide fast approximate solutions (Sanchez-Gonzalez et al., 2020; Stachenfeld et al., 2021). Such emulators are being applied in the context of weather (Pathak et al., 2022; Lam et al., 2023; Bi et al., 2023) and climate modelling (Kochkov et al., 2024; Watt-Meyer et al., 2023; Subel & Zanna, 2024), with computational speedups of up to five orders of magnitude (Kurth et al., 2023).

The standard approach to constructing a neural emulator is to predict the state of the system at some future time as a function of the current timestep. Simulated trajectories are then generated by a *rollout* of many autoregressive passes, where the predicted state is fed back into the emulator. This framework suffers from instabilities over long timescales, as accumulation of errors leads to a drift of the emulator trajectory away from the truth (Chattopadhyay et al., 2023; Bonavita, 2024; Parthipan et al., 2024; Bach et al., 2024). Eventually, the drift becomes large enough that the state is out of distribution, and trajectories exponentially diverge. Some proposed modifications to improving the stability of rollouts include the addition of noise to training data (Stachenfeld et al., 2021), training over multiple consecutive timesteps (Vlachas et al., 2020; Keisler, 2022), predicting multiple timesteps at once (Brandstetter et al., 2022), including additional input channels representing external forcings (Watt-Meyer et al., 2023; Subel & Zanna, 2024), and adding iterative refinement steps focusing on different frequency components (Lippe et al., 2023).

The situation is further complicated by the fact that the systems that are being modelled are generally chaotic. Indeed,

---

[1] Courant Institute of Mathematical Sciences, New York University, USA [2] Center for Data Science, New York University, USA. Correspondence to: Joan Bruna <bruna@cims.nyu.edu>.

*Proceedings of the 42nd International Conference on Machine Learning*, Vancouver, Canada. PMLR 267, 2025. Copyright 2025 by the author(s).

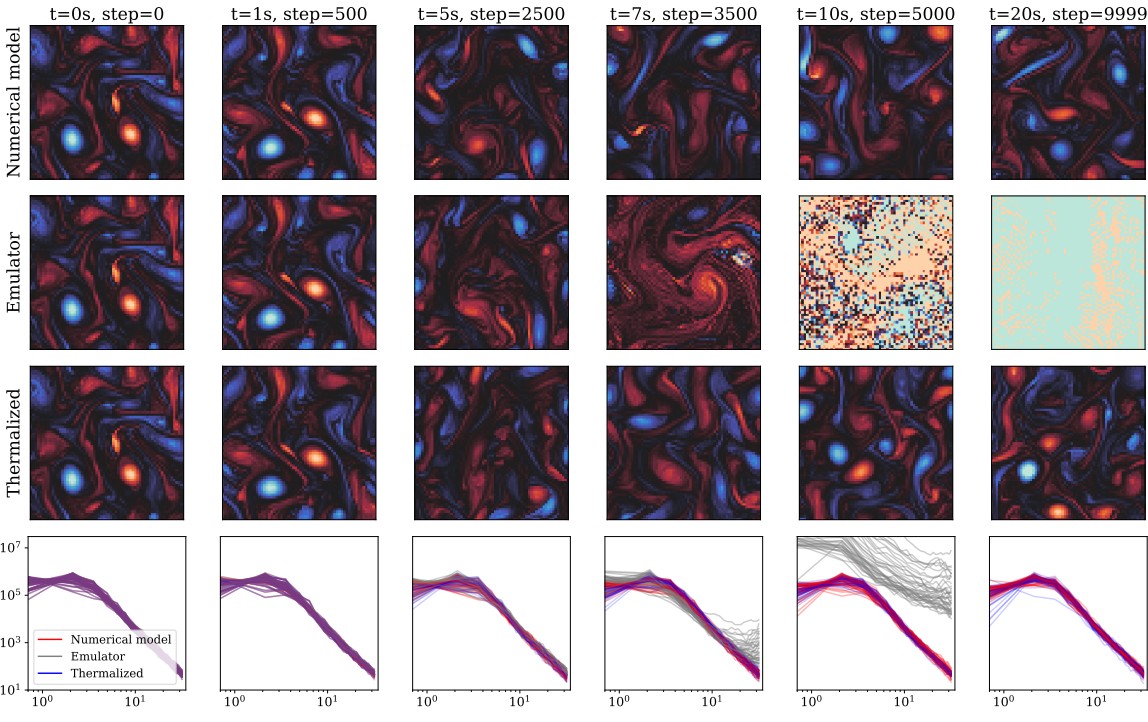

*Figure 1. Top 3 rows:* Vorticity fields from a numerical model, emulated, and thermalized trajectories for Kolmogorov flow over a trajectory of $10,000$ steps. *Bottom row:* Radially averaged kinetic energy spectra at each timestep, for $40$ randomly initialized trajectories.

for such systems with intrinsically diverging trajectories, recurrent architectures are unable to capture the dynamics while maintaining stability during training (Mikhaeil et al., 2021). Recent work has attempted to incorporate knowledge of the chaotic system's invariant measure into training to stabilize predictions and improve the modeling of chaotic dynamics (Li et al., 2022; Jiang et al., 2023; Schiff et al., 2024).

In this work, we propose an alternative approach, based on diffusion models, a framework developed for generative modeling of images (Sohl-Dickstein et al., 2015; Song & Ermon, 2019; Ho et al., 2020), to stabilize autoregressive emulator rollouts of chaotic systems. Diffusion models implement a reversible measure transport between the data distribution and a reference distribution, typically the standard Gaussian measure, by learning a one-parameter family of denoising operators, resulting in an efficient non-equilibrium sampling scheme. As errors accumulate during emulator rollouts, the state of the system moves to regions of low probability, which are then transported back to equilibrium by querying the reverse diffusion models with an appropriately tuned noise level. Importantly, the reverse diffusion defines a coupling between corrupted and equilibrium distributions that respects the temporal dynamics. This can be thought of as a form of on-the-fly denoising that prevents the

emulated trajectory from diverging away from the training distribution, thereby maintaining stability over arbitrarily long time horizons; see Figure 2. We refer to this process as *thermalization*, as the system is relaxed back towards a stationary measure. One of the key advantages of our scheme relative to prior work is the modularity of training: we only require a pretrained black box emulator and a separately trained diffusion model for the stationary measure of the dynamics. These two models are only combined at inference time, and, as shown in the numerical experiments, require only a small overhead in inference computational cost.

**Main Contributions.** We present a novel application of diffusion models in the context of using neural networks to approximate the solutions to chaotic PDEs. We demonstrate the *thermalizer* method on two high-dimensional turbulent systems and show that arbitrarily long rollouts can be achieved when an unstable neural emulator is coupled with a thermalizer.

## 2. Preliminaries

**Problem Setup.** We consider data of the form $(\mathbf{x}_t)_{t\in\mathbb{N}}$ as the time discretization of an underlying dynamical system $(\mathbf{x}_t)_{t\in\mathbb{R}_+}$, $\mathbf{x}_t \in \mathbb{R}^d$, solving $\partial_t \mathbf{x}_t = \mathcal{F}(\mathbf{x}_t)$, where $\mathcal{F}$ :

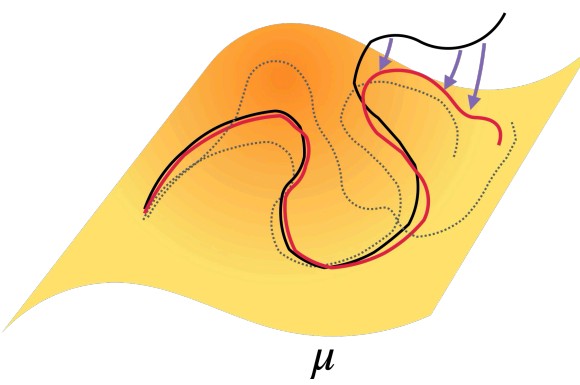

$$\mu$$

*Figure 2.* Schematic view of the thermalisation: in dashed grey, two typical trajectories from $\pi_T$, that preserve the equilibrium distribution $\mu$, here idealised in terms of its typical set. In black, an emulated trajectory, that eventually drifts away from the typical set. In purple, the reverse diffusion, which defines an approximate transport between $\hat{\mu}_t$, the law of $\mathbf{x}_t$, and $\mu$, resulting in the red 'thermalized' trajectory.

$\mathbb{R}^d \to \mathbb{R}^d$ is an unknown, time-independent, and generally non-linear operator.

To account for the uncertainty in initial conditions, the time and space discretization errors, as well as the chaotic nature of some dynamics, it is convenient to adopt a probabilistic description of the problem: we view $(\mathbf{x}_t)_{t\in\mathbb{N}}$ as a high-dimensional Markovian stochastic process $\pi$. By the time-homogeneous Markovian assumption, its joint law $\pi_t := \pi(\mathbf{x}_1, \ldots, \mathbf{x}_t)$ satisfies the recurrence equation $\pi_{t+1}(\mathbf{x}_0, \ldots, \mathbf{x}_{t+1}) = \mathcal{Q}(\mathbf{x}_{t+1}|\mathbf{x}_t)\pi_t(\mathbf{x}_0, \ldots, \mathbf{x}_t)$ for any $t > 0$, where $\mathcal{Q}$ is the Markov kernel encoding the conditional distribution of $\mathbf{x}_{t+1}$ given $\mathbf{x}_t$. We denote by $\mu_t$ the marginal distribution of $\mathbf{x}_t$. We also assume that this kernel is irreducible, which implies the existence of a unique stationary measure $\mu := \mu_\infty \in \mathcal{P}(\mathbb{R}^d)$, such that $\mu = \mu\mathcal{Q}$.

Faced with this Markov structural assumption, it is thus tempting to learn the system by focusing solely on the short-time conditional kernel $\mathcal{Q}$, given its small memory footprint, and the fact that it is sufficient to reconstruct the full joint probability distribution. In other words, leveraging the Markov kernel enables arbitrarily long generations at inference time with a constant training budget.

In this work we are particularly concerned with the long-time behavior of the system: given a possibly large time horizon $T$ (e.g., of the order of several decades in climate models compared to days for weather forecasts), our goal is to build a scalable generative model for trajectories, that is, to efficiently sample a typical trajectory from $\pi_T$, in such a way that training and inference budget remains decoupled; ie, we can produce arbitrarily long trajectories with a training cost independent of $T$. As one could anticipate, solely relying on the local Markovian structure turns out

to be insufficient in practice due to inherent instability; the key scientific question of this work is, thus, how to overcome this instability using scalable training and inference algorithms.

**Unstability of autoregressive modeling.** Given observed trajectories $\{(\mathbf{x}_1^i, \ldots, \mathbf{x}_L^i)\}_{i\le n}$, where $(\mathbf{x}_1^i, \ldots, \mathbf{x}_L^i) \sim \pi_L$ are independently drawn, one can estimate a model $\widehat{\mathcal{Q}}$ of the one-step transitions $\mathcal{Q}$. A standard choice is to assume a Gaussian transition kernel of the form $\mathcal{Q}(\cdot|\mathbf{y}) = \mathcal{N}(\Phi_\theta(\mathbf{y}), \sigma^2 I_d)$, where $\Phi_\theta$ is a generic parametric map, e.g. a neural network, so that the associated MLE corresponds to the least-square objective $\min_\theta \frac{1}{n} \sum_{i=1}^n \sum_{j<L} \|\mathbf{x}_{j+1}^i - \Phi_\theta(\mathbf{x}_j^i)\|^2$. This Gaussian model thus corresponds to a general (non-linear) diffusion in continuous time, where $d\mathbf{x}_t = \mathcal{F}(\mathbf{x}_t)dt + \sqrt{2}\sigma dB_t$, which in the limit of $\sigma = 0$ recovers deterministic dynamics. Note that, in practice, one can replace the one-step transitions with $k$-step transitions without loss of generality so that the emulated model advances $k$ times faster than the numerical timestep.

Once trained, one can sample arbitrarily long trajectories by querying $\widehat{\mathcal{Q}}$ in an autoregressive fashion: for each $T > 0$, we denote by $\widehat{\pi}_T$ the joint law of $(\mathbf{x}_0, \ldots, \mathbf{x}_T)$, where $\mathbf{x}_0 \sim \pi_0$ and $\mathbf{x}_{t+1}|\mathbf{x}_t \sim \widehat{\mathcal{Q}}(\mathbf{x}_{t+1}|\mathbf{x}_t)$. We easily verify that

$$\text{KL}(\pi_T||\widehat{\pi}_T) = \sum_{t\le T} \mathbb{E}_{\mathbf{y}\sim\mu_t}\text{KL}\left(\mathcal{Q}(\cdot|\mathbf{y})||\widehat{\mathcal{Q}}(\cdot|\mathbf{y})\right) . \quad (1)$$

For large $T$, since $\mu_t \to \mu$ by ergodicity, the above identity directly yields

$$\text{KL}(\pi_T||\widehat{\pi}_T) \simeq T\mathbb{E}_{\mathbf{y}\sim\mu}\text{KL}\left(\mathcal{Q}(\cdot|\mathbf{y})||\widehat{\mathcal{Q}}(\cdot|\mathbf{y})\right) . \quad (2)$$

This confirms the intuition that the law of the trajectories will drift apart, as measured with cross-entropy, at a rate which is linear in the horizon. Note that RHS is precisely the test error of the regression objective — in accordance with Girsanov's theorem in the continuous-time limit.

While the above KL control measures whether typical samples from $\pi_t$ are typical under our model $\hat{\pi}_t$, the generative setting, where we sample from $\hat{\pi}_t$, corresponds instead to the reverse KL divergence. Assuming that $\widehat{\mathcal{Q}}$ is also irreducible, with invariant measure $\widehat{\mu}$, inverting the roles of $\mathcal{Q}$ and $\widehat{\mathcal{Q}}$ now yields

$$\text{KL}(\widehat{\pi}_T||\pi_T) \simeq T\mathbb{E}_{\mathbf{y}\sim\widehat{\mu}}\text{KL}\left(\widehat{\mathcal{Q}}(\cdot|\mathbf{y})||\mathcal{Q}(\cdot|\mathbf{y})\right) . \quad (3)$$

Crucially, we now have a shift between the test error distribution $\widehat{\mu}$ and the training distribution $\mu$, leading to a dramatic impact in the generalisation guarantees of the model and consistent with the observed instability of emulator rollouts. Note though that the ergodicity assumption on $\hat{\mathcal{Q}}$ is merely for convenience: in practice, it may be that $\hat{\mathcal{Q}}$ is not irreducible, e.g., when $\sigma^2 = 0$, failing to converge towards

a stationary measure, or, using again the continuous-time formulation, when the associated Fokker-Plank equation $\partial_t \mu_t = \nabla \cdot (-\mathcal{F}\mu_t) + \Delta\mu_t$ admits no equilibrium, due to the fact that the irrotational component of $\mathcal{F}$ does not have enough decay at infinity.

**Mitigating distribution shifts.** In the face of the previous discussion, it is thus tempting to control the extent of distribution shift by adding a regularisation term in the training objective. Given a model $\mathcal{Q}_\theta$ for the one-step transitions, and assuming again irreducibility, it has an associated invariant measure $\mu_\theta$, characterized as the (unique) Perron eigenfunction of $\mathcal{Q}_\theta$. Provided one can access (or estimate) the true invariant measure $\mu$ of the system, one can thus consider a learning objective that combines both sources of error:

$$L(\theta) = \mathbb{E}_{\mathbf{y}\sim\mu}\mathrm{D}(\mathcal{Q}(\cdot|\mathbf{y}), \mathcal{Q}_\theta(\cdot|\mathbf{y})) + \lambda\tilde{\mathrm{D}}(\mu, \mu_\theta). \quad (4)$$

Here, we keep the presentation informal, and consider generic (and possibly distinct) probability metrics $\mathrm{D}, \tilde{\mathrm{D}}$, as long as they admit Monte-Carlo estimators. The important aspect of Eq (4), however, is the presence of $\mu_\theta$, which is only known implicitly as the invariant measure of the kernel $\mathcal{Q}_\theta$. While some prior works, e.g. (Schiff et al., 2024), have explored such learning objective in physical applications, it presents an important computational challenge, since it requires estimating the invariant measure $\mu_\theta$ each time $\theta$ is updated. In challenging situations where $\mathcal{Q}_\theta$ does not have a substantial spectral gap, obtaining $\mu_\theta$ from $\mathcal{Q}_\theta$ may become prohibitively expensive. In the next section, we introduce an alternative framework to address such distribution shift, which also relies on having a model for the invariant measure $\mu$ of the true system but crucially avoids the computation of $\mu_\theta$.

**Estimating the Invariant Measure using Diffusion Models.** Diffusion models (Sohl-Dickstein et al., 2015; Song & Ermon, 2019; Ho et al., 2020) have been tremendously successful in the field of generative modeling. The framework is based on modeling the inverse of an iterative noising process, which is best thought of as the discretisation of an underlying continuous diffusion process. Let us recall the main ideas which will be needed to define our method.

Given $X_0 \sim \nu_0 := \mu$, we consider the Ornstein-Ulhenbeck (OU) process $dX_s = -X_s ds + \sqrt{2}dB_s$, where $B_s$ is the standard Brownian motion. The law $\nu_s$ of $X_s$ solves the associated Fokker-Plank equation $\partial_s \nu_s = \nabla \cdot (x\nu_s) + \Delta\nu_s$, which can also be written as the transport equation $\partial_s \nu_s = \nabla \cdot ((x + \nabla \log \nu_s)\nu_s)$. It is well known that the law $\nu_S$ converges exponentially fast (in KL) to the standard Gaussian measure $\gamma_d := \mathcal{N}(0, I_d)$, so $\nu_S \approx \gamma_d$ for $S$ large enough. Now, by changing the sign of the velocity field and introducing again the diffusive Laplacian term, this

transport equation can be formally reversed: if $\tilde{\nu}_s$ solves $\partial_s \tilde{\nu}_s = \nabla \cdot ((-x - 2\nabla \log \nu_{S-s})\tilde{\nu}_s) + \Delta\tilde{\nu}_s$, with $\tilde{\nu}_0 = \nu_S$, then $\tilde{\nu}_S = \nu_0$. In other words, the reverse diffusion

$$d\tilde{X}_s = (\tilde{X}_s + 2\nabla \log \nu_{S-s}(\tilde{X}_s))ds + \sqrt{2}dB_s, \quad (5)$$

with $\tilde{X}_0 \sim \gamma_d$, satisfies $\tilde{X}_S \sim \nu_0$ up to an exponentially small error (due to the difference $\mathrm{KL}(\nu_S || \gamma_d) = O(e^{-S})$).

The above procedure defines a non-equilibrium sampling scheme that requires access to the scores $\nabla \log \nu_s$ along the OU semigroup. As it turns out, such scores can be efficiently estimated from original samples of $\mu$ via a denoising objective. Indeed, defining the Denoising Oracle $D(\mathbf{y}, \sigma) := \mathbb{E}[\mathbf{x}|\mathbf{y}]$, where $\mathbf{y} = \mathbf{x} + \sigma\mathbf{z}$, $\mathbf{x} \sim \mu$ and $\mathbf{z} \sim \gamma_d$, Tweedie's formula (Robbins, 1992), relates the score to the Denoising oracle via an explicit affine relationship, $\nabla \log \nu_s(\mathbf{x}) = -\alpha_s^{-1}(\mathbf{x} + \beta_s D(\mathbf{x}, \alpha_s))$ for explicit values of $\alpha_s$ and $\beta_s$. Finally, $D(\cdot, \sigma)$ can be efficiently estimated from samples using the MSE variational characterisation of the posterior mean: $D = \arg\min_{F:\mathbb{R}^d \times \mathbb{R} \to \mathbb{R}^d} \int \mathbb{E}_{\mathbf{x}\sim\nu, \mathbf{z}\sim\mathcal{N}(0,I)}[\|X - F(X + \sigma Z)\|^2]d\sigma$. Here $\alpha_s$ and $\beta_s$ are as defined in (Ho et al., 2020).

The standard Denoising Diffusion Probabilistic Model (DDPM) framework (Ho et al., 2020) introduced an efficient implementation of the above scheme, by considering appropriate discretization of the noise levels, and where the denoiser $D = \mathbf{D}_\phi$ is represented by a neural network with parameters $\phi$. In the generative modeling case, the noise scheduling spans the full range from the unperturbed data distribution at $s = 0$, to pure Gaussian noise at $s = S$ (where $S$ is generally in the range $\approx 1000$). As we shall see next, we will introduce a different use of diffusion models. Using the available data, we can estimate a diffusion model for the equilibrium distribution $\mu$, that we will use to build a coupling between $\hat{\mu}$, a perturbation of the equilibrium, and $\mu$. This coupling can be interleaved with timestepping performed by the emulator, constraining the emulated trajectory to stay in regions of state space consistent with the training data, a process we call *thermalization*. Compared to the generative modeling approach, we are only interested in small amounts of denoising at the very low noise end of the scheduler, i.e. at low $s$.

## 3. Related work

**Large-scale autoregressive models:** In the context of climate and weather modelling, several large scale autoregressive models have been developed (Pathak et al., 2022; Lam et al., 2023; Bi et al., 2023; Lang et al., 2024; Kochkov et al., 2024), and are indeed being deployed for commercial and public use. Despite the tremendous success of these models over short timescales, long-term instability due to error accumulation over large numbers of timesteps is still a common problem.

**Diffusion models for PDEs:** In (Lippe et al., 2023), an autoregressive model is used to predict the next timestep, and then a small number of "refinement" steps are applied to the prediction. These refinement steps are trained on a denoising loss, and can therefore be interpreted as DDPM denoising steps, analogous to our thermalization steps. Therefore our work is similar in nature, however with the key differences that their denoiser is a conditional model, and that our denoising model is constructed separately to the emulator and applied adaptively at inference time. Additionally, we investigate rollouts over many more autoregressive passes, up to $1e^5$ steps. Diffusion models have also been applied to generating realizations of turbulent fields (Lienen et al., 2023), and emulation in the form of autoregressive conditional models (Kohl et al., 2023; Price et al., 2023), where the probabilistic nature of predictions is able to improve modeling of chaotic dynamics. To reduce the computational cost of generating next step predictions, (Gao et al., 2024) performs the conditional generation in a learned, low-dimensional latent space, and (Shehata et al., 2025) develops a more efficient conditional generation algorithm. Diffusion models have also been applied to large-scale weather modelling, both for the purposes of emulation (Price et al., 2023; Rühling Cachay et al., 2024) and for statistical downscaling (Mardani et al., 2023; Wan et al., 2023).

**Stabilizing rollouts:** A central problem in fully learned autoregressive simulators is the accumulation of errors and resultant instabilities. Several regularisation terms have been proposed (Chattopadhyay et al., 2023; Schiff et al., 2024; Guan et al., 2024). (Brandstetter et al., 2022) used training over multiple consecutive timesteps to improve stability, while the addition of training noise has also been demonstrated to improve stability when using a single timestep prediction (Stachenfeld et al., 2021). The study in (List et al., 2024) investigated an approach to multi-timestep training without propagating gradients. The inclusion of forcing from external systems has been shown to stabilize extremely long rollouts in emulation of ocean models (Watt-Meyer et al., 2023; Subel & Zanna, 2024). Decomposing the system into linear and non-linear components was also demonstrated to improve stability in simple 1D non-linear systems (Linot et al., 2023).

## 4. Methodology

### 4.1. Emulator

We first describe the framework for the construction of the emulator. We represent the neural-network model as $\Phi_\beta$. We found the best performance when using an emulator to model the residuals between two timesteps, i.e., a predicted state can be denoted $\hat{\mathbf{x}}_{t+1} = \hat{\mathbf{x}}_t + \Phi_\beta(\hat{\mathbf{x}}_t) + \tau\mathbf{n}$ where the final term represents the stochastic component, so $\mathbf{n} \sim \mathcal{N}(0, \mathbf{I})$ and we set $\tau = 1e^{-5}$ throughout the paper. The

neural network parameters $\beta$ are optimized by minimizing a MSE loss function:

$$\mathcal{L}_\beta = \frac{1}{N} \sum_{k=1}^{N} \sum_{t=0}^{L-1} \parallel \Phi_\beta(\hat{\mathbf{x}}_t^k) - (\mathbf{x}_{t+1}^k - \mathbf{x}_t^k) + \tau\mathbf{n} \parallel^2, \quad (6)$$

where trajectories start from a simulation snapshot ($\hat{\mathbf{x}}_0 = \mathbf{x}_0$), $L$ represents the length of a training trajectory (throughout this paper, we use $L = 4$), $k$ indexes an individual training trajectory, and the loss is averaged over a total of $N$ training trajectories. Gradients are backpropagated through the full $L$ timesteps, as done in (Brandstetter et al., 2022; List et al., 2024). We note that the separation between training snapshots, $\Delta t$, is in general larger than the numerical timestep. The neural network $\Phi_\beta$ is implemented using two choices: a U-Net (Unet) style architecture (Ronneberger et al., 2015), where we set the number of filters and downsampling layers such that the model has 48M parameters, and a Dilated ResNet (DRN) (Stachenfeld et al., 2021) with 0.5M parameters. Further details on architecture and optimization can be found in Appendix B. To build a training and test set, we use numerical simulations to generate a total of $N = 500,000$ trajectories. We use $450,000$ of these for training and the remaining $50,000$ for validation and testing. The resulting network thus defines a point-estimate, associated with the local transition kernel $\widehat{\mathcal{Q}}(\mathbf{x}|\mathbf{y}) = \delta(\mathbf{x} - (\mathbf{y} + \Phi_\beta(\mathbf{y})))$, a degenerate version of the Gaussian transition kernel discussed in Section 2. Additionally, we implement a stochastic emulator by including small amounts of noise at each timestep, during both training and inference. The solution manifold for dynamical systems is commonly a fractal, leading to a singular invariant measure. The problem of learning such a solution manifold can be facilitated by applying small perturbations to the training data (Zeeman, 1988; Gritsun & Branstator, 2007; Baldovin et al., 2022). Note that this is naturally achieved in the case of the diffusion model via use of the denoising score loss.

### 4.2. Thermalizer

We now introduce a procedure to stabilize trajectories that operates at inference time, assuming a pretrained model for local dynamics $\widehat{\mathcal{Q}}$ given by the previous emulator model, and a pretrained diffusion model for $\mu$, consisting of the denoiser $\mathbf{D}_\phi$.

For each time $t$, let $\hat{\mu}_t$ denote the law of the current generated field $\mathbf{x}_t$. Unstability is to be expected as soon as this law drifts sufficiently far away from $\mu$, that we recall corresponds to the distribution of $\mathbf{y}$ used to train the model for local dynamics $\mathcal{Q}(\cdot|\mathbf{y})$. For instance, one can use the 2-Wasserstein distance $W_2(\hat{\mu}_t, \mu)$ to quantify this drift. This distance is interesting because it also prescribes an *optimal* coupling $\Gamma_t$ between $\hat{\mu}_t$ and $\mu$ minimizing the transportation cost: $W_2^2(\hat{\mu}_t, \mu) = \mathbb{E}_{(\mathbf{x},\mathbf{y})\sim\Gamma_t}\|\mathbf{x} - \mathbf{y}\|^2 =$

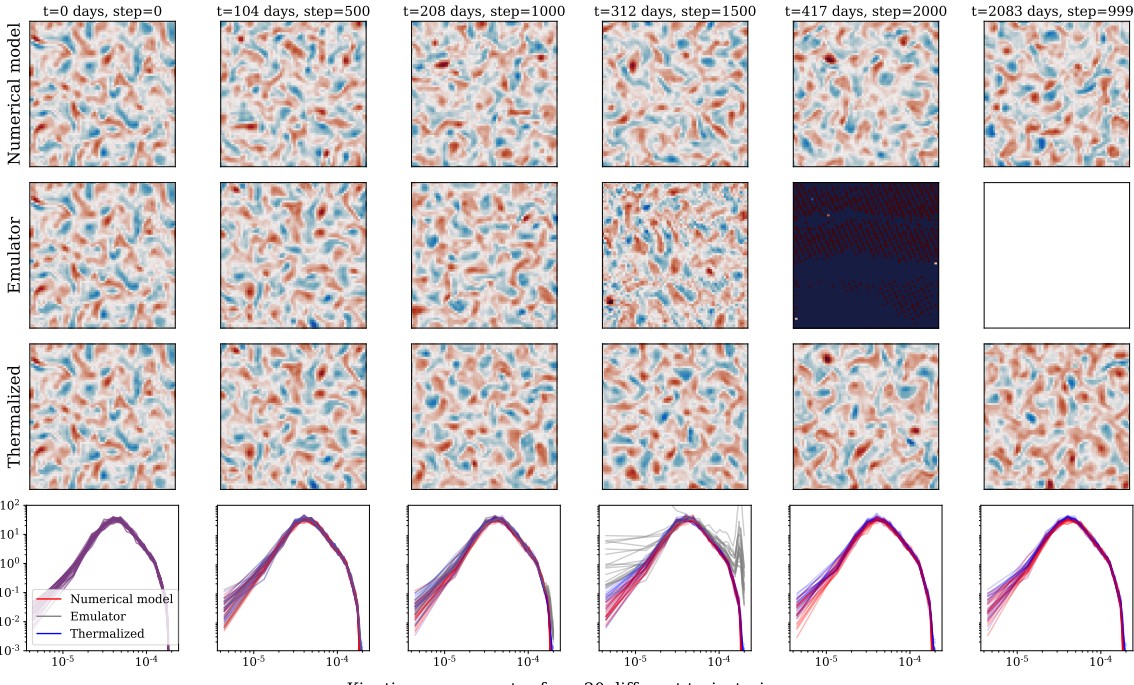

*Figure 3. Top 3 rows:* QG trajectories over $10,000$ steps for the numerical model (top), neural network emulator (second row), and the thermalized neural emulator (third row). We show the potential vorticity, $q$, in the top layer of the 2-layer QG system. *Bottom row:* Kinetic energy spectra for each timestep, for 20 different trajectories. We show the layer thickness-weighted average of the spectra in both the upper and lower layers.

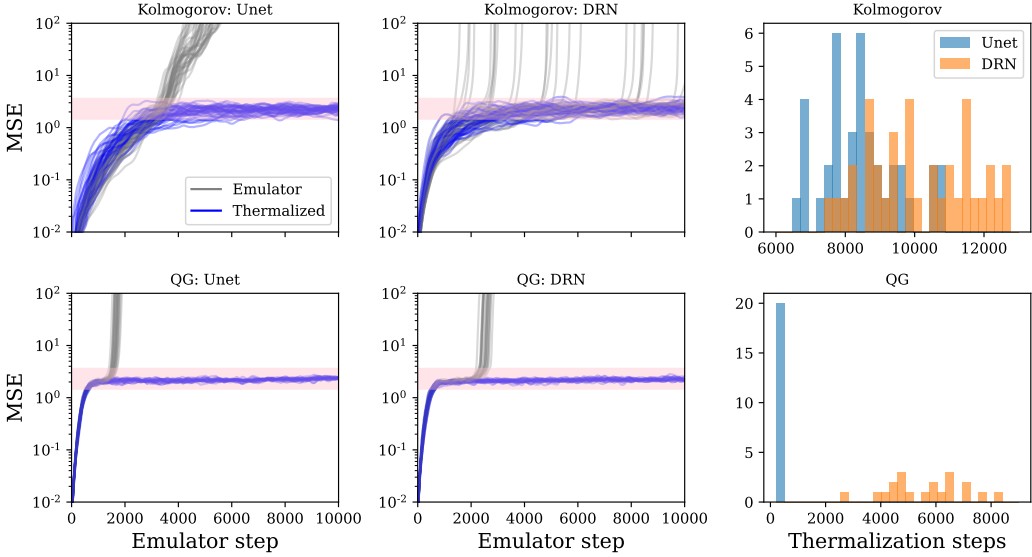

*Figure 4.* Mean squared error between neural network trajectories and numerical model trajectories for different baseline architectures (Unet and Dilated ResNet), starting from the same initial conditions. We show 40 Kolmogorov flow trajectories in the top two panels, and 20 QG flows in the lower two panels. Thermalized flows remain around a constant MSE long after non-thermalized flows have become unstable. The region where state vectors are decorrelated in time, but stable, is shown in shaded pink. On the right, we show histograms for the total number of thermalization steps along the full trajectories.

$\min_{\Gamma \in \Pi(\hat{\mu}_t, \mu)} \mathbb{E}_{(\mathbf{x}, \mathbf{y}) \sim \Gamma} \|\mathbf{x} - \mathbf{y}\|^2$, where we recall that a coupling $\Gamma \in \Pi(\mu_1, \mu_2)$ is a joint distribution in $\mathbb{R}^d \times \mathbb{R}^d$ such that its marginals are respectively $\mu_1$ and $\mu_2$. Moreover, under mild assumptions Chewi et al. (2024, Section 1.4), this optimal coupling is associated with an optimal transport map $\mathcal{T}_t : \mathbb{R}^d \to \mathbb{R}^d$, such that $(\mathbf{x}, \mathcal{T}_t(\mathbf{x})) \sim \Gamma_t$. In other words, the current field $\mathbf{x}_t \sim \hat{\mu}_t$ can be mapped 'back' to the equilibrium distribution via $\mathcal{T}_t(\mathbf{x}_t)$, and this map minimizes the expected distortion $\|\mathbf{x}_t - \mathcal{T}_t(\mathbf{x}_t)\|^2$ introduced in the trajectory.

This defines an idealised stabilization method, which unsurprisingly is unfeasible to implement. First, it rests on the ability to estimate the law $\hat{\mu}_t$ *from a single realization* $\mathbf{x}_t$. While we have samples from $\mu$ in the training set, generating independent samples from $\hat{\mu}_t$ requires running the emulator multiple times at inference, which can be quickly prohibitively expensive. Even then, it is well-known that optimal transport suffers from the curse of dimensionality Weed & Bach (2019).

Instead, we will leverage the diffusion model to provide an efficient approximation to these optimal transport maps. Our approximated transport will consist of running the reverse diffusion (5) starting at a noise level $s = s(t)$, that we adjust as a function of $\mathbf{x}_t$, and $X_s = \mathbf{x}_t$, resulting in the coupling $(X_0, X_s) \sim \widehat{\Gamma}_t$. The rationale behind this choice of transport map is the following: one can view the samples $\mathbf{x}_t \sim \hat{\mu}_t$ as a perturbation of 'correct' equilibrium states coming from $\mu$; if these perturbations were isotropic Gaussian, then the law $\hat{\mu}_t$ would agree with $\nu_s$ for some noise level $s = s(t)$, and the reverse diffusion would implement a valid transport towards $\mu$. While this transport is *not* the optimal transport, one has some control over the suboptimality gap Albergo et al. (2023b, Prop 3.1).

Such guarantees are no longer valid for generic, non-Gaussian perturbations. Instead, if one assumes an ideal choice of the noise level, given by $s^*(t) = \arg\min_s \mathrm{KL}(\hat{\mu}_t \| \nu_s)$, and $\tilde{\mu}_t$ is the law of $X_0$ conditioned on $X_{s^*} = \mathbf{x}_t \sim \hat{\mu}_t$, by the data-processing inequality one has

$$\mathrm{KL}(\tilde{\mu}_t \| \mu) \leq \mathrm{KL}(\hat{\mu}_t \| \nu_{s^*}) \leq \mathrm{KL}(\hat{\mu}_t \| \nu_0) = \mathrm{KL}(\hat{\mu}_t \| \mu) . \tag{7}$$

Thus, assuming that the noise level is correctly estimated, and an accurate denoising model, the thermalizer step contracts towards equilibrium. Moreover, for nearly Gaussian perturbations, the LHS can be substantially smaller than the original error. It is important to emphasize that even approximate transports can go a long way to stabilize rollouts, as long as they can mitigate the distribution shift.

It is interesting to compare this scheme with the *Langevin map*, that runs the Fokker-Plank equation $\partial_t \nu_t = \nabla \cdot ((\nabla \log \mu) \nu_t) + \Delta \nu_t$ starting from $\nu_0 = \hat{\mu}$. While the

---

**Algorithm 1** Algorithm for thermalized trajectories

> **for** $t = 1$ **to** $N$ **do**
> $\quad \mathbf{n} \sim \mathcal{N}(0, \mathbf{I})$
> $\quad \mathbf{x}_t = \mathbf{x}_{t-1} + \Phi_\beta(\mathbf{x}_{t-1}) + \tau \mathbf{n}$
> $\quad s_{\mathrm{pred}} = \arg\max \mathbf{D}_\phi^{(2)}(\mathbf{x}_t)$
> $\quad$ **if** $s_{\mathrm{pred}} > s_{\mathrm{init}}$ **then**
> $\quad\quad \epsilon \sim \mathcal{N}(0, \mathbf{I})$
> $\quad\quad \mathbf{x}_t = \sqrt{\bar{\alpha}_{s_{\mathrm{pred}}}} \mathbf{x}_t + \sqrt{1 - \bar{\alpha}_{s_{\mathrm{pred}}}} \epsilon$
> $\quad\quad$ **for** $s = s_{\mathrm{pred}}$ **to** $s_{\mathrm{stop}}$ **do**
> $\quad\quad\quad \mathbf{z} \sim \mathcal{N}(0, \mathbf{I})$
> $\quad\quad\quad \mathbf{x}_t = \frac{1}{\sqrt{\alpha_s}} \left( \mathbf{x}_t - \frac{1-\alpha_s}{\sqrt{1-\bar{\alpha}_s}} \mathbf{D}_\phi^{(1)}(\mathbf{x}_t) \right) + \sqrt{\beta_s} \mathbf{z}$
> $\quad\quad$ **end for**
> $\quad$ **end if**
> **end for**

---

Langevin map indeed maps any initial measure to $\mu$ as $t \to \infty$, it suffers from two important limitations: first, its time to relaxation can be exponentially large in presence of meta-stability and multiple modes, even for initial distributions $\hat{\mu}$ close to equilibrium, affecting the resulting transportation cost. Next, while the score $\nabla \log \mu$ corresponds to the zero-noise limit of the DDPM objective, this score is only accurately learnt nearby the data distribution $\nu$, and we found it unstable for our purposes. One can view our proposed transport scheme as a non-equilibrium counterpart of the Langevin map that provides improved robustness.

It remains to be discussed how to find a suitable noise level $s(t)$, given the current emulator state. To tackle this question, we modify the standard implementation of the DDPM framework, by adding a classifier output such that the noise level $s$ is *predicted* by the network, instead of being passed as an input parameter. We denote these two outputs using a superscript, where $\mathbf{D}_\phi^{(1)}(\mathbf{x}_t) \in \mathbb{R}^d$ represents the denoised field, and $\mathbf{D}_\phi^{(2)}(\mathbf{x}_t) \in \mathbb{R}^S$ represents the predicted categorical distribution over $S$ noise levels. The details of how this additional classifier head are added to the Unet are given in Appendix C. The loss function we minimise is

$$\mathcal{L}_\phi = \frac{1}{N} \sum_{i=1}^{N} \Big[ \| s_i \epsilon_i - \mathbf{D}_\phi^{(1)}(\mathbf{x}_i + s_i \epsilon_i) \|^2$$

$$+ \sum_{s=1}^{S} y_s \log \mathbf{D}_\phi^{(2)}(\mathbf{x}_i + s_i \epsilon_i) \Big] , \tag{8}$$

where $s$ is uniformly sampled from $s \in [1, S]$, $\mathbf{x}_t$ is the training data, and $y_s \in \mathbb{R}^S$ is a one-hot vector encoding the true noise level. To implement the thermalizer, we use the same Unet architecture as for the emulator, where the network now has 53M parameters, with the additional parameters constituting the noise-classifying head. To train the thermalizer, we use the same $N = 450,000$ training set

as for the emulator, except we take only the first snapshot of each short trajectory to avoid training the thermalizer on redundant data samples.

During an emulator rollout, at each timestep we run a forward pass of the classifier-component of the network to predict the noise level, such that for a given state vector $\mathbf{x}$, the predicted noise level is $s_{\text{pred}} = \arg\max \mathbf{D}_\phi^{(2)}(\mathbf{x})$. If the predicted noise level exceeds a threshold, which we call $s_{\text{init}}$, we then apply $s_{\text{therm}} = s_{\text{pred}} - s_{\text{stop}}$ steps of denoising to the state vector, where $s_{\text{pred}}$ is the estimated noise level in the state vector, and $s_{\text{stop}}$ is some minimum acceptable noise level, below which we do not thermalize. The steps for this process are detailed in Algorithm 1.

## 5. Experiments

Additional visualizations including videos are available here.

### 5.1. Kolmogorov flow

To test our framework on turbulent flows, we use a fluid flow described by 2D incompressible Navier-Stokes equations with sinusoidal forcing, often referred to as Kolmogorov flow, used in many works on DL for turbulence modelling (Boffetta & Ecke, 2012; Kochkov et al., 2021; Lippe et al., 2023; Schiff et al., 2024). The baseline emulator is constructed using the loss in equation 6, where we set $\Delta t = 2\delta t$, where $\delta t$ is the numerical model timestep, and the number of recurrent passes used during training $L = 4$.

In Figure 1 we compare flow trajectories for three models: in the top row, a direct numerical simulation, in the second row, a neural network emulator constructed as outlined in section 4.1, and in the third row, a flow trajectory using the same emulator, but including applications of the thermalizer model described in 4.2. Each trajectory is initialized at a state obtained from the numerical model. Whilst initially, the flow fields are all consistent after $\sim 3500$ passes, errors begin to accumulate in the emulator field, and instabilities become visible. By 5000 steps, the instabilities dominate and the trajectory has entirely diverged. The thermalized flow, however, remains stable and consistent with the numerical model, as a result of the small corrections applied dynamically during the emulator rollout. In the lowest row of Figure 1, we show the kinetic energy spectra at each timestep, for 40 different flow trajectories, demonstrating that the stabilizing effect of the thermalizer is consistent across all randomly initialized trajectories.

### 5.2. Quasi-geostrophic turbulence

As an additional test case, we repeat the experiment on a 2-layer quasi-geostrophic (QG) turbulent system. This system

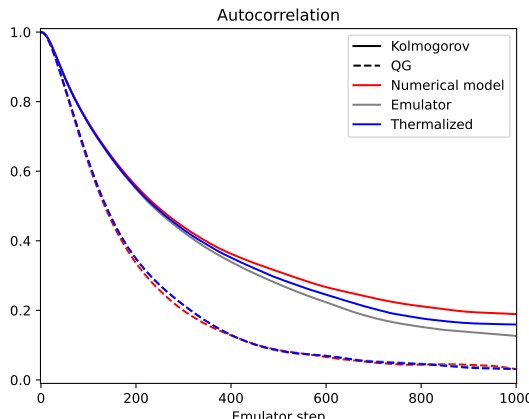

*Figure 5.* Autocorrelation over time for numerical model, and neural network trajectories.

represents a stratified fluid in a rotating frame. This kind of turbulent dynamics is prevalent in climate science, and plays a central role in both atmospheric and ocean dynamics (Majda & Qi, 2018). The dataset construction procedure is the same as with Kolmogorov flow, using dataset sizes of $N = 500,000$ fluid snapshots for both the emulator and thermalizer, with the only change being that now a fluid snapshot has 2 layers, and the emulator training timestep is decreased to $\Delta t = 5\delta t$. The model architectures are kept the same, except with an additional input channel to accommodate the additional fluid layer. When training over $L = 4$ timesteps, we found that the Unet emulator trajectories went unstable in the region of $\sim 2e^4$ to $5e^4$ timesteps, so to bring the emulator in line with our other baselines, we reduced the number of training timesteps to $L = 2$. For the DRN, we keep $L = 4$ as with the Kolmogorov emulators. Details on the numerical solver and configuration of the QG system are given in Appendix A.2.

In Figure 3, we compare trajectories for the numerical model, emulator and thermalized trajectories. As with the Kolmogorov flow, after approximately 4000 timesteps the un-thermalized trajectories accumulate sufficient error to diverge, but the thermalized flows remain stable. In the lower panel we show the radially averaged kinetic energy spectra for 20 trajectories at each timestep. This is the weighted average of the kinetic energy spectra in the upper and lower layers, weighted by the layer thickness.

In Figure 4, we show the MSE with respect to the numerical model, for different neural network models and flow configurations. We observe that initially the MSE increases as trajectories diverge over the first few hundred steps, and settle to approximately unity MSE. In this regime, shaded in pink, the state vectors are stable but decorrelated. States with MSE$\gg 1$ are unstable. Over $\approx 2000$ timesteps emulator trajectories all go unstable (with the exception of the

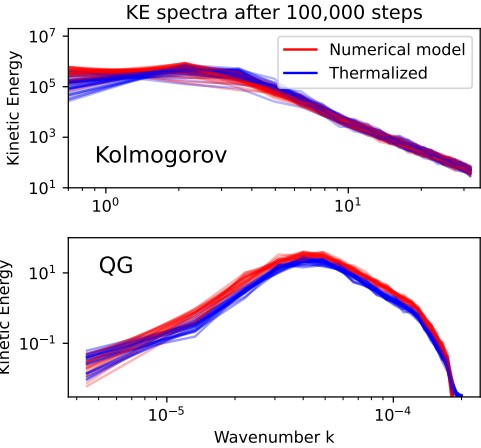

*Figure 6.* Radially-averaged kinetic energy for Kolmogorov (top) and QG (bottom) trajectories from a thermalized Unet emulator after $1e^5$ timesteps.

Kolmogorov DRN, where many trajectories stay stable for much longer). However all thermalized trajectories stay stable for the full $10,000$ steps, due to the corrective steps applied during the rollout. In the right panels, we show the number of thermalization steps applied along the entire trajectory for all models and flow configurations. In the case of the QG Unet, the thermalization is highly efficient, with just $\sim 140$ thermalization steps required on average to stabilise each $10,000$ step trajectory. In terms of computational cost, we run a forward pass of the noise-classifying head of the thermalizer at every timestep, which is $60\%$ the cost of a Unet emulator step, as we are only activating the downsampling and noise-classifying components of the equivalent-sized thermalizer Unet. Additional forward passes of the full thermalizer network are only run in the case where $s_{\mathrm{pred}} > s_{\mathrm{init}}$, so given an accurate noise classifier, the algorithm is inherently efficient and only applies corrective denoising steps when necessary.

Given the fact that thermalizing steps induce some small distortion in the field, it is important to consider the effect on the temporal consistency of the flow trajectories. In Figure 5, we show the autocorrelation for both Kolmogorov and QG fields, averaged across all 40 and 20 trajectories respectively. We see no significant change in the autocorrelation between the numerical model, and the thermalized flow, indicating that the temporal disruption caused by the thermalization steps is minimal. Finally, in figure 6, we show kinetic energy spectra for Kolmogorov and QG thermalized flows after $1e^5$ steps, where the spectra demonstrate that all trajectories are successfully stabilised over this long rollout. However we note that there is some offset in the KE spectra of the QG fields, which show consistently less kinetic energy than in the numerical model.

## 6. Conclusion

We introduced the *thermalizer*, an algorithm for stabilising autoregressive surrogate models leveraging a pretrained diffusion model of the stationary data distribution. This diffusion model provides a family of transport maps from Gaussian perturbations of the stationary measure back to its equilibrium, which are shown to be robust to non-Gaussian errors. As a result, this diffusion model can be deployed at inference time to constrain trajectories to stay along the solution manifold, mitigating the accumulation of error due to autoregression of an approximate timestepping model. An appealing aspect of the model is the lightweight training: we treat the emulator as a pre-trained black-box, and train the diffusion model of the equilibrium separately, using standard pipelines.

A crucial component of the thermalizer algorithm is the adaptive denoising steps, which serve to minimize both the computational cost of the algorithm, and disruption of the temporal dynamics of the flow trajectories. We demonstrate this approach on two high-dimensional turbulent systems, a forced 2D Navier-Stokes flow, and a 2-layer quasi-geostrophic turbulent flow, enabling stable predictions over $1e^5$ emulator steps.

**Limitations and Future Work.** The main limitation of our current framework is the underlying (isotropic) Gaussianity assumption on the perturbations. Even though our experiments demonstrate some form of robustness to such misspecification, this comes at the expense of a careful choice of hyperparameters. In particular, performance is strongly dependent on the setting of $s_{\mathrm{init}}$ and $s_{\mathrm{stop}}$, which must be individually tuned for each combination of emulator and thermalizer via brute force search. Also indeed some configurations remain stable, however with imperfect kinetic energy spectra at large timestep, as shown in Figure 6. We suspect that such limitations could be addressed by replacing Gaussian diffusion models with a base measure more adapted to the errors introduced by the emulator; stochastic interpolants *aka* flow matching (Albergo et al., 2023a; Lipman et al., 2022) provide such flexibility. In that respect, an interesting theoretical question is to understand the robustness of the OU-based coupling to non-Gaussian perturbations, beyond the data-processing inequality. Next, we focused in the time-homogeneous setting, owing to its importance in applications and the particularly lightweight implementation of the thermalizer. The next natural step is to consider autonomous systems, eg with a forcing term having some temporal periodicity. Finally, another important future direction is to consider other domains where long autoregressive rollouts are often used — perhaps even LLMs?

## Acknowledgements

The authors would like to thank Fabrizio Falasca, Pavel Perezhogin, Stephane Mallat, Etienne Lempereur, Freddy Bouchet, Eric Vanden-Eijnden and Edouard Oyallon for many valuable discussions. This project is supported by Schmidt Sciences, LLC.

## Impact Statement

This paper presents work whose goal is to advance the field of Machine Learning. There are many potential societal consequences of our work, none of which we feel must be specifically highlighted here.

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

# A. Dynamical systems

We evaluate our method on two high-dimensional dynamical systems, exhibiting the kinds of chaotic, turbulent dynamics that are prevalent in systems of practical interest.

## A.1. Kolmogorov flow

A common fluid flow to test ML surrogates, is to use a forced variant of incompressible Navier-Stokes:

$$\partial_t \mathbf{u} + \nabla \cdot (\mathbf{u} \otimes \mathbf{u}) = \nu \nabla^2 \mathbf{u} - \frac{1}{\rho} \nabla p + \mathbf{f} \tag{9}$$

where $\mathbf{u} = [u_x, u_y]$, $\nu$ is the kinematic viscosity, $\rho$ is the fluid density, $p$ is the pressure, and $\mathbf{f}$ is an external forcing term. Following (Kochkov et al., 2021), we use a constant sinusoidal forcing function. We use $p = 1$ and $\nu = 0.001$, corresponding to a Reynolds number $\mathrm{Re} = 10,000$.

We numerically solve the equations using the pseudospectral method with periodic boundary conditions from the publicly available code `jax-cfd` (Dresdner et al., 2022). We use a numerical timestep of $\delta t = 0.001$s, and simulate the system with $n_x = n_y = 512$ spatial gridsteps. The fields are then downsampled to $64 \times 64$ for training and testing of the emulator. We use the vorticity, $\omega = \nabla \times \mathbf{u} \in \mathbb{R}^{64 \times 64}$ as our state vector, unlike in some previous works which trained their emulators on the velocity vector fields (e.g. Lippe et al. (2023)). All state vectors are normalized to unit variance before passing to the neural network, and results in the main text are shown in normalised units for simplicity.

## A.2. Quasi-geostrophic turbulence

We consider a two-layer, quasi-geostrophic system, where the prognostic variable is the potential vorticity, given by

$$q_m = \nabla^2 \psi_m + (-1)^m \frac{f_0^2}{g' H_m} (\psi_1 - \psi_2), m \in \{1, 2\}, \tag{10}$$

where $m = 1$ denotes the upper layer, $m = 2$ denotes the lower layer, $H_m$ is the depth of the layer, $\psi$ is the streamfunction, which is related to the fluid velocity by $\mathbf{u}_m = (u_m, v_m) = (-\partial_y \psi_m, \partial_x \psi_m)$, and $f_0$ is the Coriolis frequency. The time evolution of the system is given by

$$\partial_t q_m + \nabla \cdot (\mathbf{u}_m q_m) + \beta_m \partial_x \psi_m + U_m \partial_x q_m = -\delta_{m,2} r_{ek} \nabla^2 \psi_m + \mathrm{ssd} \circ q_m, \tag{11}$$

where $U_m$ is the mean flow in the $x$ (zonal) direction, $\beta_m = \beta + (-1)^{m+1} \frac{f_0^2}{g' H_m}(U_1 - U_2)$, $r_{ek}$ is the bottom drag coefficient, and $\delta_{m,2}$ is the Kronecker delta.

We numerically solve these equations using a pseudo-spectral method, with an Adams-Bashforth 3rd order timestepper in the Fourier domain and periodic boundary conditions. This was implemented in PyTorch, the code for which will be made publicly available upon de-anonymization. Numerical simulations are run at a resolution of $256 \times 256$, and the potential vorticity fields are downsampled to $64 \times 64$ for training and testing of the neural network models. We use the 2-layer potential vorticity, $q \in \mathbb{R}^{2 \times 64 \times 64}$ as our state vector representing a QG system, so all neural networks used in QG experiments have an additional input channel compared to the architectures used in Kolmogorov flow. As with the Kolmogorov experiments, all state vectors are normalized to unit variance before passing to the neural network, and results in the main text are shown in normalised units for simplicity.

# B. Baseline emulator

## B.1. Loss function

Here we describe the construction of the baseline neural emulator. Training over multiple timesteps has been comprehensively shown to improve long term stability (Brandstetter et al., 2022; Gupta & Brandstetter, 2023; List et al., 2024; 2025), and one can imagine three different ways to build a neural network and loss function to do this. First, we can simply predict the state at some future timestep, as a function of the current timestep, i.e. $\Phi_\beta : \mathbf{x}_t \to \mathbf{x}_{t+1}$, with MSE loss:

$$\mathcal{L}_\beta = \frac{1}{N} \sum_{k=1}^{N} \sum_{t=0}^{L-1} \| \Phi_\beta(\hat{\mathbf{x}}_t^k) - \mathbf{x}_{t+1}^k + \tau \mathbf{n} \|^2 \tag{12}$$

where $k$ indexes a training trajectory (of a total of $N$ training trajectories, each of length $L$ snapshots), $\hat{\mathbf{x}}_{t+1} = \Phi_\beta(\hat{\mathbf{x}}_t) + \tau \mathbf{n}$ denotes a predicted state (including stochastic component), and $\hat{\mathbf{x}}_0 = \mathbf{x}_0$ such that we are starting from a simulation snapshot. An alternative approach is to instead emulate the residual between two snapshots, i.e., $\hat{\mathbf{x}}_{t+1} = \hat{\mathbf{x}}_t + \Phi_\beta(\hat{\mathbf{x}}_t) + \tau \mathbf{n}$. This can be achieved using two loss functions - either by evaluating the loss on the predicted state as in equation 12:

$$\mathcal{L}_\beta = \frac{1}{N} \sum_{k=1}^{N} \sum_{t=0}^{L-1} \| \left( \Phi_\beta(\hat{\mathbf{x}}_t^k) + \hat{\mathbf{x}}_t^k \right) - \mathbf{x}_{t+1}^k + \tau \mathbf{n} \|^2 \tag{13}$$

where all terms are the same as previous. Alternatively, one can evaluate the loss on the residual:

$$\mathcal{L}_\beta = \frac{1}{N} \sum_{k=1}^{N} \sum_{t=0}^{L-1} \| \Phi_\beta(\hat{\mathbf{x}}_t^k) - (\mathbf{x}_{t+1}^k - \mathbf{x}_t^k) + \tau \mathbf{n} \|^2 . \tag{14}$$

After experimenting with all 3 loss functions, we found significantly better performance from the residual emulators, and particularly from the residual loss function in equation 14, in terms of validation loss and MSE accuracy over 200 step rollouts.

### B.2. Architecture

We experiment with two architectures for the underlying emulator. First, we use a Unet (Ronneberger et al., 2015) style image-to-image architecture, adapted from `https://github.com/pdearena/pdearena` (Gupta & Brandstetter, 2023) (referred to as `Modern Unet` in that work). Our Unet configuration consists of 3 downsampling layers, where the number of convolutional filters is doubled at each stage of downsampling. We set the number of convolutional filters in the initial layer to 64, which leads to a total number of 48M parameters. We include residual connections between each stage of downsampling, and use GeLU activations throughout the architecture. We experimented with and without batch and group normalization, and found that these had no impact on emulator performance, so we use no normalization in the results presented in this paper. Given that both of the PDE solution data we are training and testing on were generated with periodic boundary conditions, all convolutional layers use circular padding.

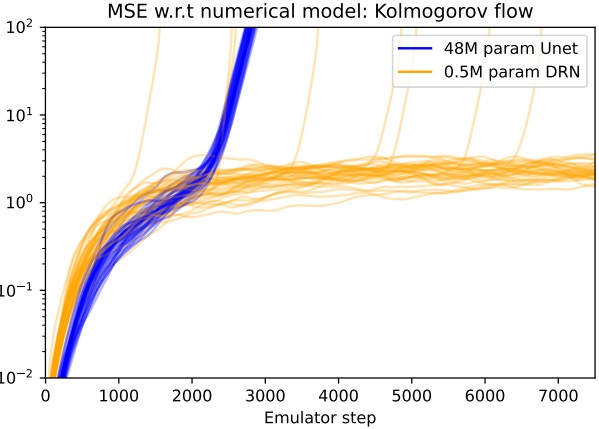

*Figure 7.* MSE with respect to a numerical model for emulated Kolmogorov flow trajectories for the Unet and DRN models. The DRN exhibits better long-term stability than the Unet, despite the much smaller model having a higher initial MSE.

Secondly, we also use a Dilated ResNet (DRN) architecture (Stachenfeld et al., 2021). Our implementation for the DRN is again adapted from (Gupta & Brandstetter, 2023). A DRN is composed of Dilated Blocks, each with 2 stacks of 7 consecutive convolutional layers. The filters of these convolutional layers have dilations of $[1, 2, 4, 8, 4, 2, 1]$ consecutively. Our baseline DRN is composed of 2 layers of convolutional preprocessing layers with stride 1, and then 4 Dilated Blocks, before finally passing through 2 final convolutional layers, again with stride 1. After each layer, we use GeLU activations, and we include residual connections over all Dilated Blocks. Each convolutional layer consists of 32 convolutional filters, and we do not use any batch or group normalization. The total number of parameters for the DRN is 0.5M.

### B.3. Optimization

The network weights are optimized using the `AdamW` optimizer with momenta $\beta_1 = 0.9$ and $\beta_2 = 0.999$, and a learning rate of $5e^{-4}$. In practice, equation 6 is broken up into mini-batches of size 32. We train the model for 12 epochs, (longer training runs were experimented but we found the loss curves did not improve with more epochs). The time horizon $\Delta t = 2$ was chosen by experimenting with a range of different values in the range $[1, 20]$, and selecting $\Delta t$ based on emulator performance in terms of MSE predictions over a time horizon of 200 step rollouts. Consistent with previous works, we found that composing multiple passes of a network trained over shorter training time horizons produced better MSE than training over larger timesteps (Lippe et al., 2023).

Importantly, we observed that improvements in validation loss often had little impact on long term stability. This can be seen in Figure 7, where we compare the short term MSE for the Unet and DRN models. The larger Unet model has significantly lower MSE over a short number of steps, but all trajectories quickly go unstable. Indeed, we observed that long term stability varied significantly with network weight initialization, even for a fixed dataset, model architecture and optimization algorithm.

## C. Diffusion model thermalizer

### C.1. Architecture

To implement the thermalizer as a diffusion model, we use the same core architecture as the Unet used for the emulator. As discussed in section 4.2, a key component of the thermalizer is some prediction of the noise level. To augment this standard Unet implementation with a noise-classifying head, we take the lowest-dimensional representation from the Unet, and add an additional 2 convolutional layers, with kernel size of 3 and stride of 1. We then vectorize the output of the last convolutional filter into a length 4096 vector. This is then passed through 2 more linear layers of size 1000. This final vector of length 1000 represents the predicted categorical distribution over noise levels for the input image. The total number of learnable parameters for this network is 53M. We implement a method to run a forward pass through the noise-classifying component of the network only, for computational efficiency during rollouts. Also note that since our network does not include the noise level as an input scalar, our diffusion model does not need have additional timestep embedding components.

### C.2. Optimization

To optimise the network, equation 8 is broken up into mini-batches of size 64. Again we use the `AdamW` optimizer, with a learning rate of $2e^{-5}$, and train the thermalizer for 35 epochs. Given that at inference time, we generally only use low noise levels of approximately $s \lesssim 20$, we experimented with training the model only on these low noise levels. However we found that performance degraded significantly, so sample noise levels $s$ uniformly across the full range during training. Noise levels are set using a cosine variance scheduler. All model architecture, training and inference codes can be found at `https://github.com/Chris-Pedersen/thermalizer`.

### C.3. Thermalizer settings: $s_{\text{init}}$ and $s_{\text{stop}}$

A key component of the algorithm is the adaptive nature of thermalization - which allows for stabilisation with a minimal interruption of temporal dynamics. There are two free parameters here - $s_{\text{init}}$ which determines the estimated noise level at which the thermalizer starts to apply corrections to the state vector, and $s_{\text{stop}}$, which sets the lowest noise level to which we run the denoising process. To find the optimum settings, we run a simple grid search on $s_{\text{init}} \in [10, 6]$ and $s_{\text{stop}} \in [5, 2]$ (note that we need $s_{\text{start}} > s_{\text{stop}}$. For each pair of $s_{\text{init}}$ and $s_{\text{stop}}$, we run a thermalized trajectory for $1e4$ steps. We chose the combination of $s_{\text{init}}$ and $s_{\text{stop}}$ in which the kinetic energy spectra of the thermalized trajectories best matches the kinetic energy in the numerical model. This experiment is repeated independently for each flow configuration, and each emulator - so this procedure was run a total of 4 times. We found $s_{\text{init}} = 7$ and $s_{\text{stop}} = 4$ optimal for the Kolmogorov Unet, $s_{\text{init}} = 10$ and $s_{\text{stop}} = 5$ optimal for the Kolmogorov DRN. For QG we use $s_{\text{init}} = 9$ and $s_{\text{stop}} = 4$ for the Unet, and $s_{\text{init}} = 6$ and $s_{\text{stop}} = 3$ for the DRN emulator.

## D. Inspecting a single thermalization step

It is possible to investigate the effect of a single thermalization step on a rollout. We randomly select a snapshot from the *thermalized* trajectories show in Figure 3, and take emulator steps from this snapshot until the predicted noise level reaches

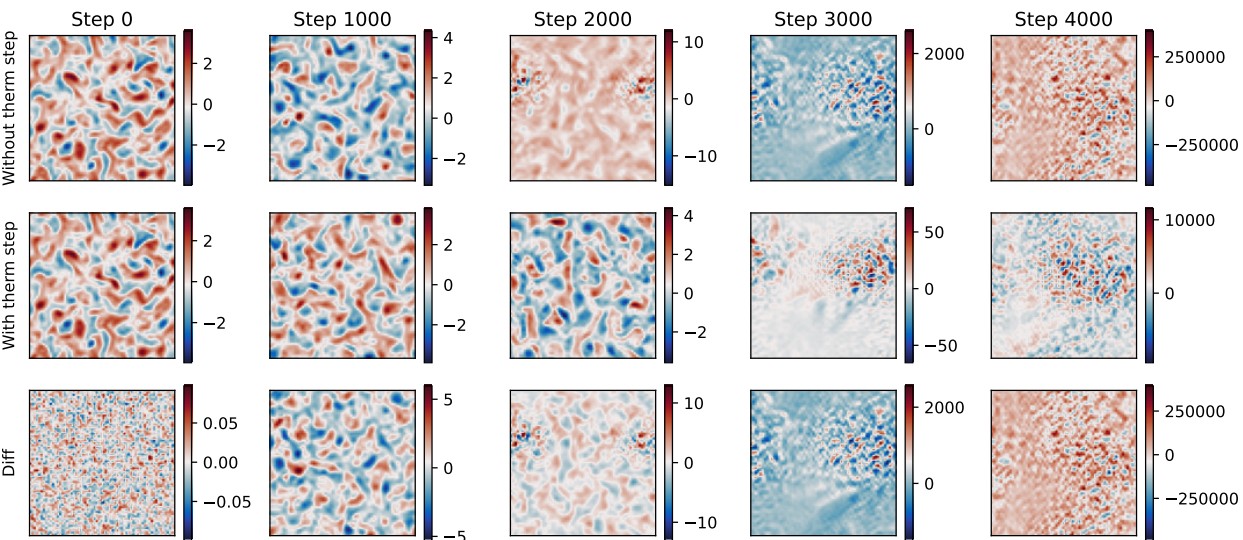

*Figure 8.* Inspecting the effect of a single thermalization step on the stability of a QG rollout. Top row shows an emulator rollout starting from a snapshot at noise level $s_{\text{pred}} = 10$. The middle row shows the same rollout, except starting from the same snapshot after applying a thermalization step. The development of instability is delayed by this single thermalization step.

the $s_{\text{init}} = 10$. We then take $s_{\text{therm}} = s_{\text{init}} - s_{\text{stop}} = 4$ thermalizing steps on this snapshot. The corresponding fields before and after thermalizing are shown in the leftmost panel of Figure 8, where the total effect of the thermalization step is shown in the lowest left panel.

We can then continue running the emulator on the snapshots with (middle row) and without (top row) applying this single thermalization step, and observe that the eventual instability develops sooner in the non-thermalized (top) row. This can be seen both visually in the potential vorticity fields as accumulation of high-frequency signal, and the growing field magnitudes shown by the colorbars. This experiment illustrates the effect that the subtle thermalization steps have on the stability of rollouts. The same procedure is repeated for other randomly selected snapshots in Figures 9 and 10.

## E. Additional results and figures

In Figure 11 we show a schematic illustrating the principle of the thermalizer. In Figure 12, we show $s_{\text{pred}}$ along the full 10000 step trajectory, for the Unet emulator in both Kolmogorov and QG flows. In gray lines we show $s_{\text{pred}}$ for the emulator trajectories, thermalized trajectories are shown in blue, and the numerical model in red. Initially, $s_{\text{pred}}$ is low for all three different models, as the state vector has not had a chance to accumulate significant error. We see that $s_{\text{pred}}$ increases as the un-thermalized trajectories accumulate error - eventually going unstable, at which point our noise-classifying output becomes unreliable, leading to the vertical gray lines. In contrast, the blue lines are constrained to stay at a low noise level, never exceeding $s_{\text{pred}} = 16$. This is the consequence of the thermalizer keeping states in-distribution. Finally the red lines for the numerical model are shown as a sanity check that our noise-classifying head is able to consistently identify that the numerical model states are all in-distribution.

In Figure 13, we inspect the number of thermalization steps along a 500 step segment of the trajectories show in Figures 1 and 15. For each of the 40 trajectories, and each of the 500 steps between timesteps $5,000$ and $5,500$, we show the number of thermalization steps applied to the state vector. We see that the amount of thermalization applies varies strongly along a trajectory, with steps ranging from 0 to 16. This indicates that the flow oscillates between periods of no correction, and periods where significant correction is applied.

Next we show the vorticity fields for 25 random samples from the 40 trajectories shown in Figure 1, at the end of the $10,000$ steps. In In Figure 16, we show results for the numerical model. Above each panel, we show the predicted noise level for each flow state. In Figures 17 and 18, we show the equivalent for the emulator and thermalized trajectories. Note that in the emulator case, after $10,000$ steps the fields have gone so far out of distribution that the predicted noise levels are unreliable.

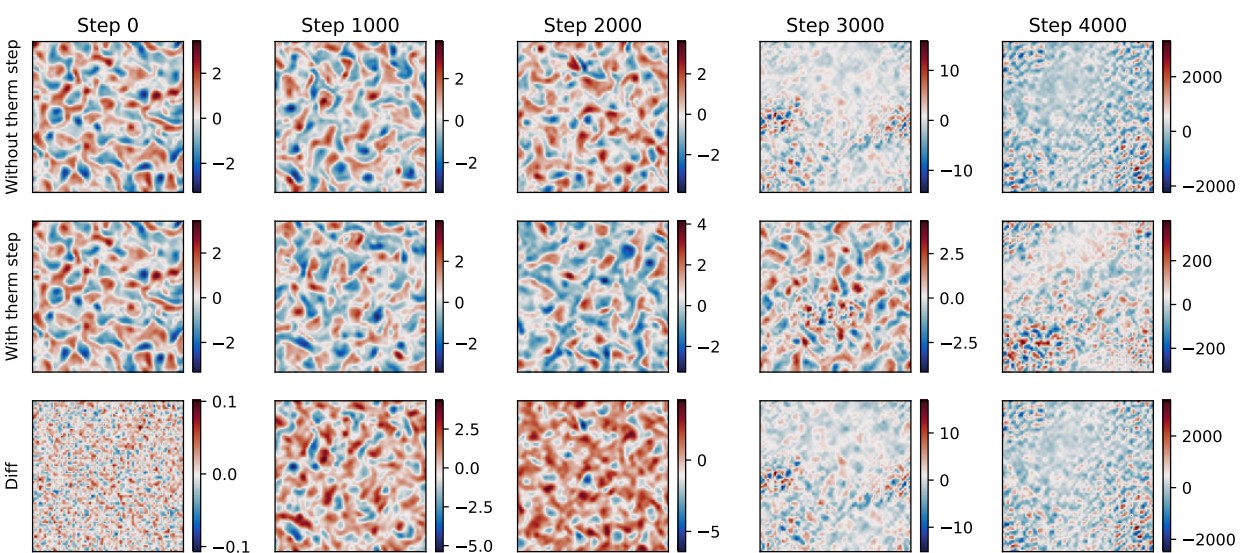

*Figure 9.* Same as Figure 8, except with a different initial condition.

In Figures 19, 20 and 21, we show the upper layer potential vorticity for all 20 QG trajectories after $10,000$ steps, for the numerical, emulator, and thermalized models respectively.

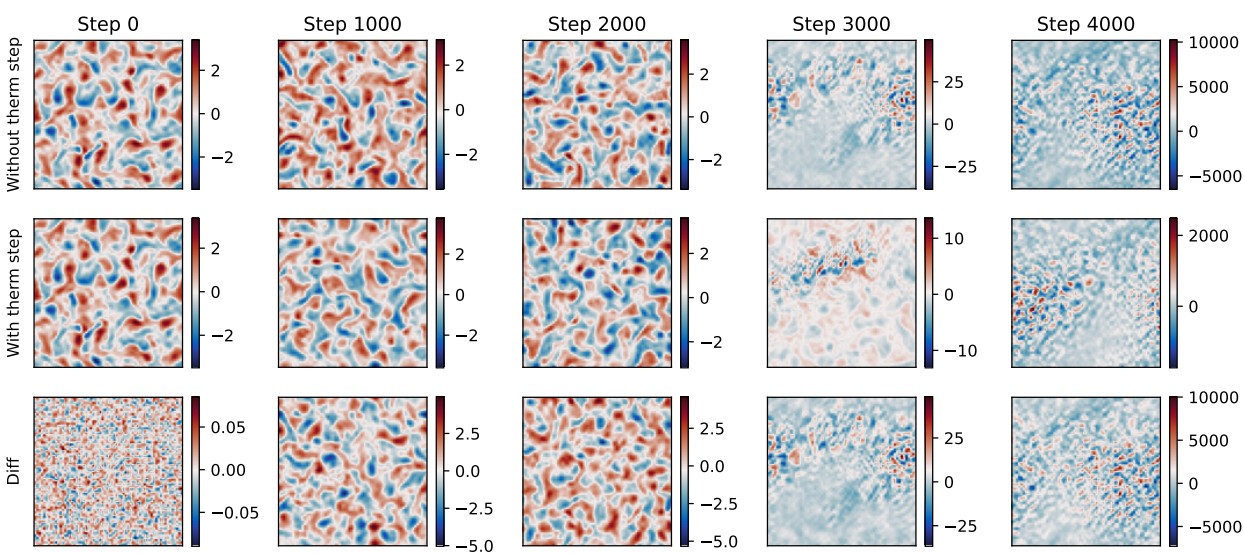

*Figure 10.* Same as Figures 8 and 9, except with a different initial condition.

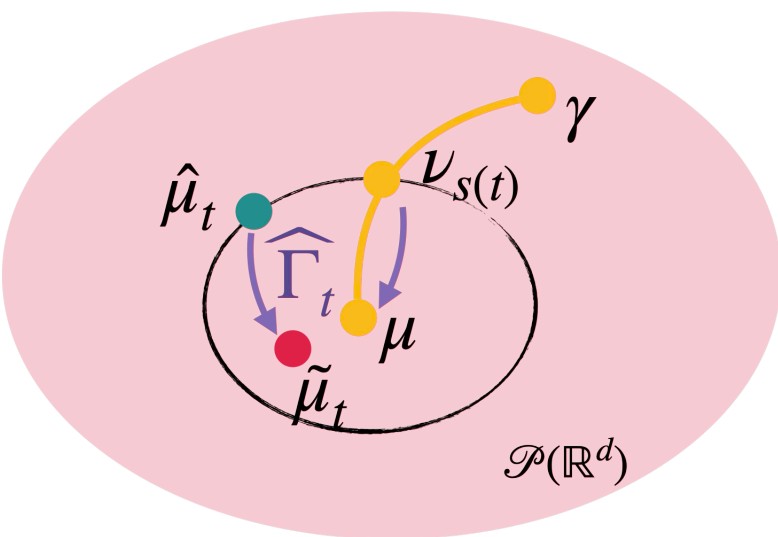

*Figure 11.* Schema for the thermalization in the space of marginal distributions $\mathcal{P}(\mathbb{R}^d)$. In yellow, the OU diffusion path is defining a transport between the equilibrium $\mu$ and the Gaussian measure $\gamma$. In green, the current law $\hat{\mu}_t$ of $\mathbf{x}_t$. In purple, the (stochastic) transport map $\hat{\Gamma}_t$ defined by the reverse diffusion. While by construction it transports $\nu_s$ back to $\mu$, when applied to $\hat{\mu}_t$ it produces the corrected measure $\tilde{\mu}_t$, reducing the error as per (7).

Predicted noise level during rollouts

Kolmogorov

QG

*Figure 12.* Predicted noise level, $s_{\mathrm{pred}}$ during rollouts for the numerical model, emulator, and thermalized trajectories. In the top panel we show results for the 40 Kolmogorov trajectories using the Unet emulator shown in figure 1, and in the bottom panel we show the 20 QG trajectories shown in figure 3.

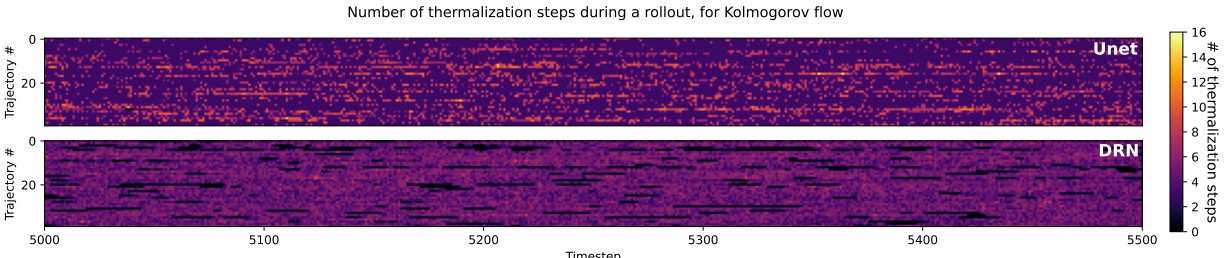

Number of thermalization steps during a rollout, for Kolmogorov flow

*Figure 13.* Number of thermalization steps along thermalized trajectories. We inspect the number of thermalization steps for each of the 40 trajectories at each timestep, between timestep 5000 and 5500, for both the Unet and DRN emulators on Kolmogorov flow.

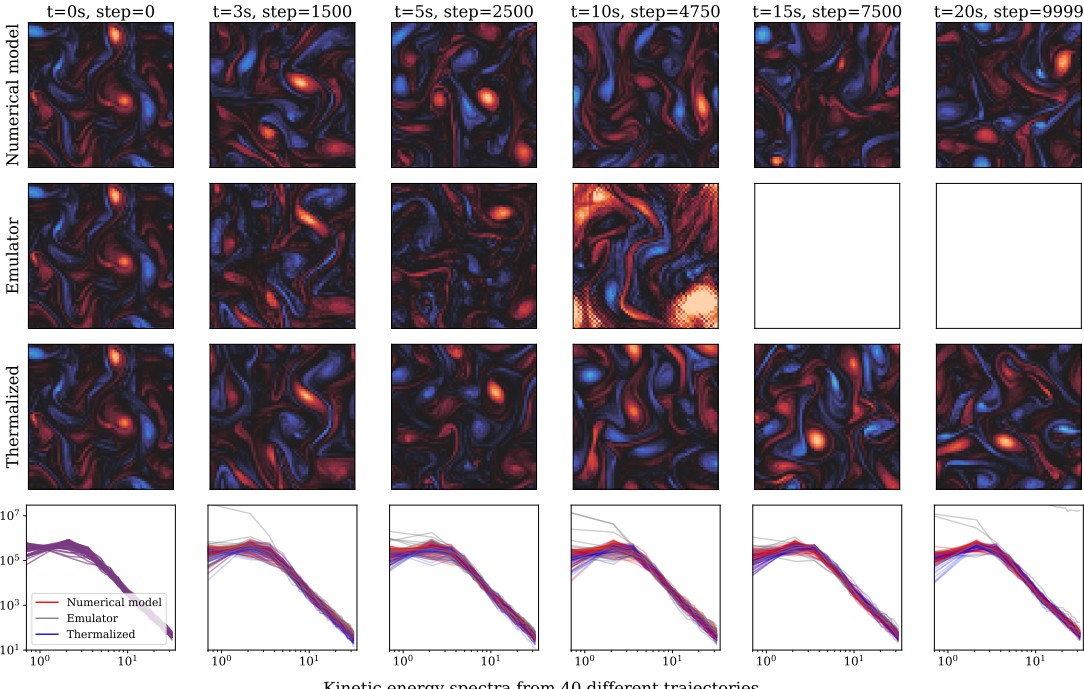

*Figure 14.* Equivalent figure as in Figure 1, but with the Dilated ResNet (DRN) emulator.

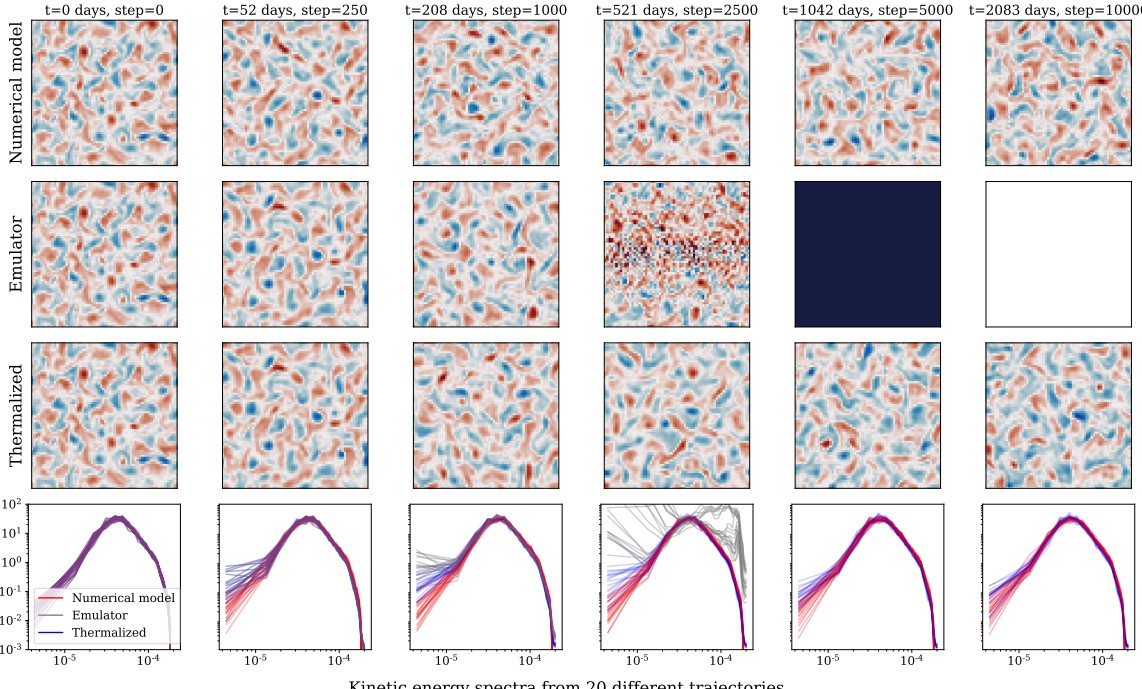

*Figure 15.* Equivalent figure as in Figure 3, but with the Dilated ResNet (DRN) emulator.

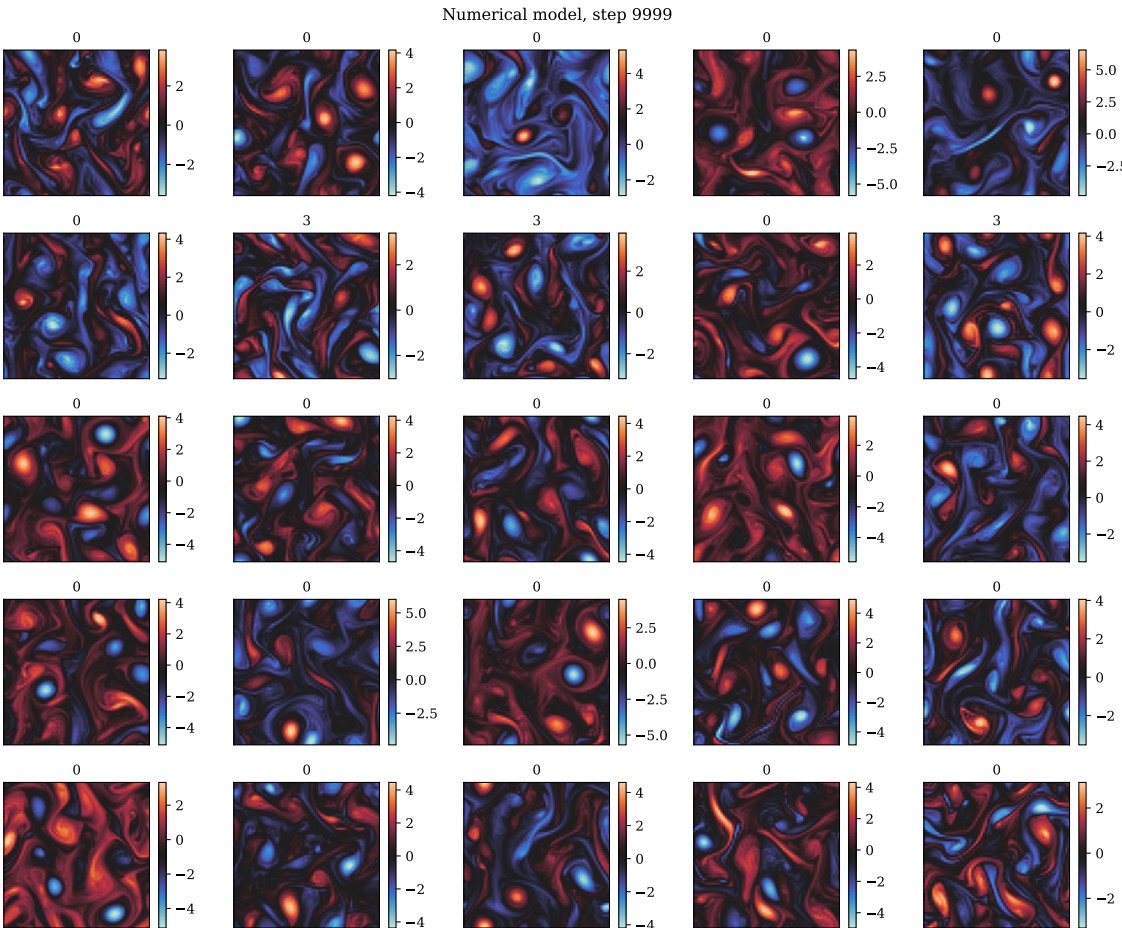

*Figure 16.* Kolmogorov vorticity snapshots after $10,000$ steps, from the numerical model. The predicted noise level is shown above each fluid state.

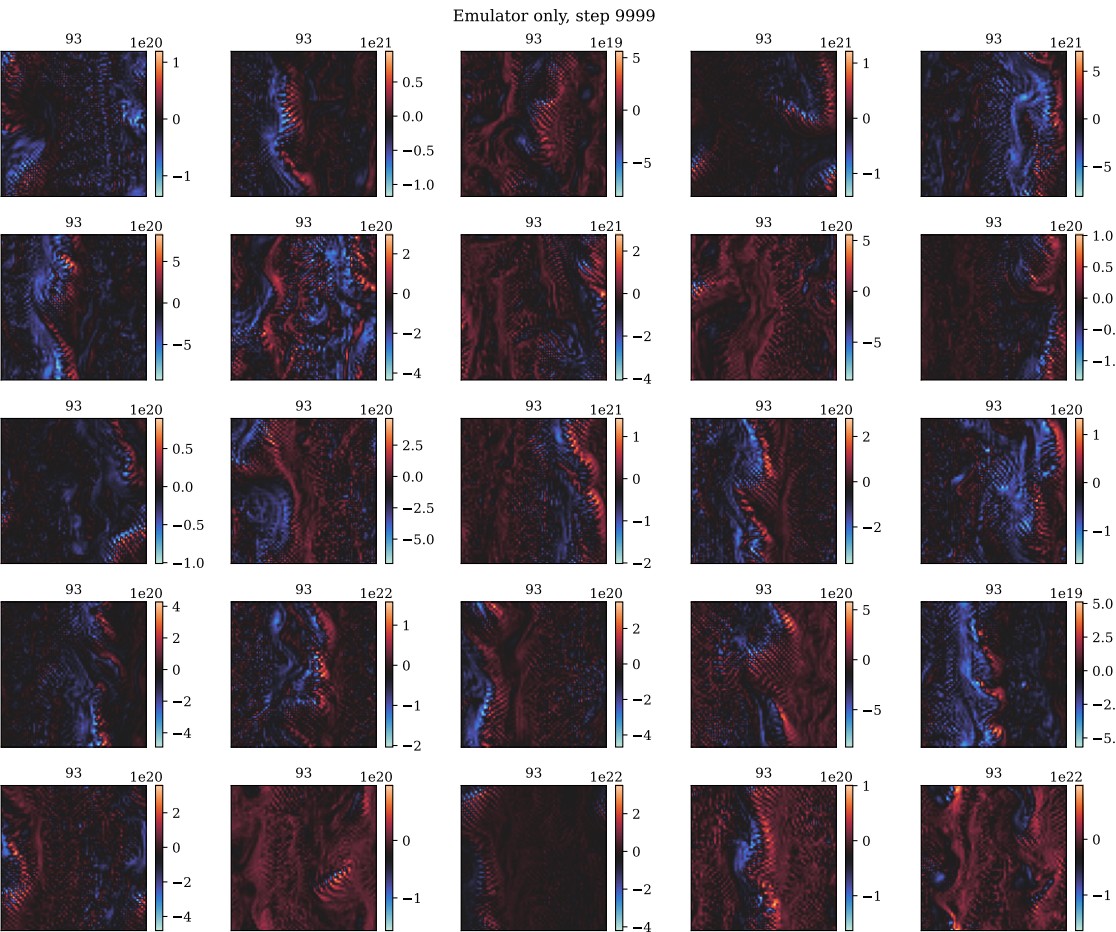

*Figure 17.* Kolmogorov vorticity snapshots after 10,000 steps, from the neural network emulator. The predicted noise level is shown above each fluid state.

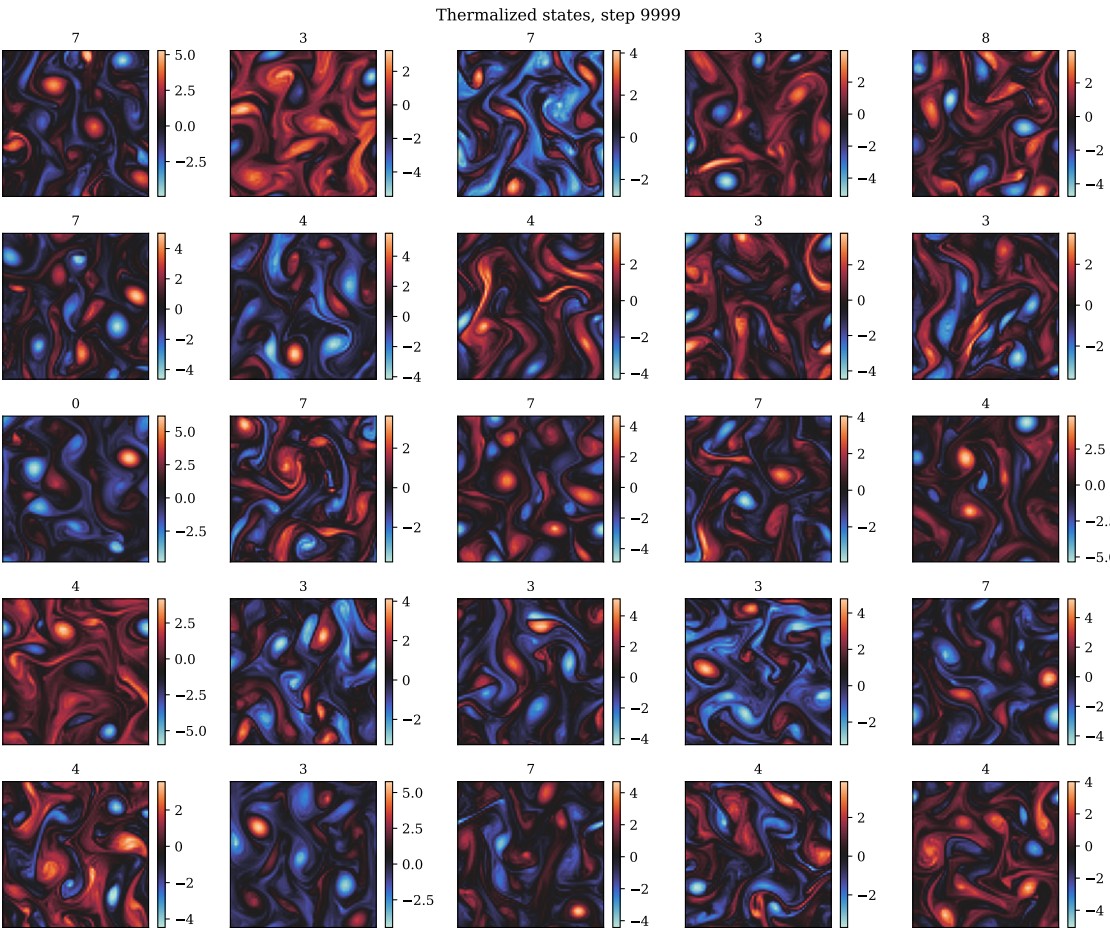

*Figure 18.* Kolmogorov vorticity snapshots after $10,000$ steps, for the thermalized trajectories. The predicted noise level is shown above each fluid state.

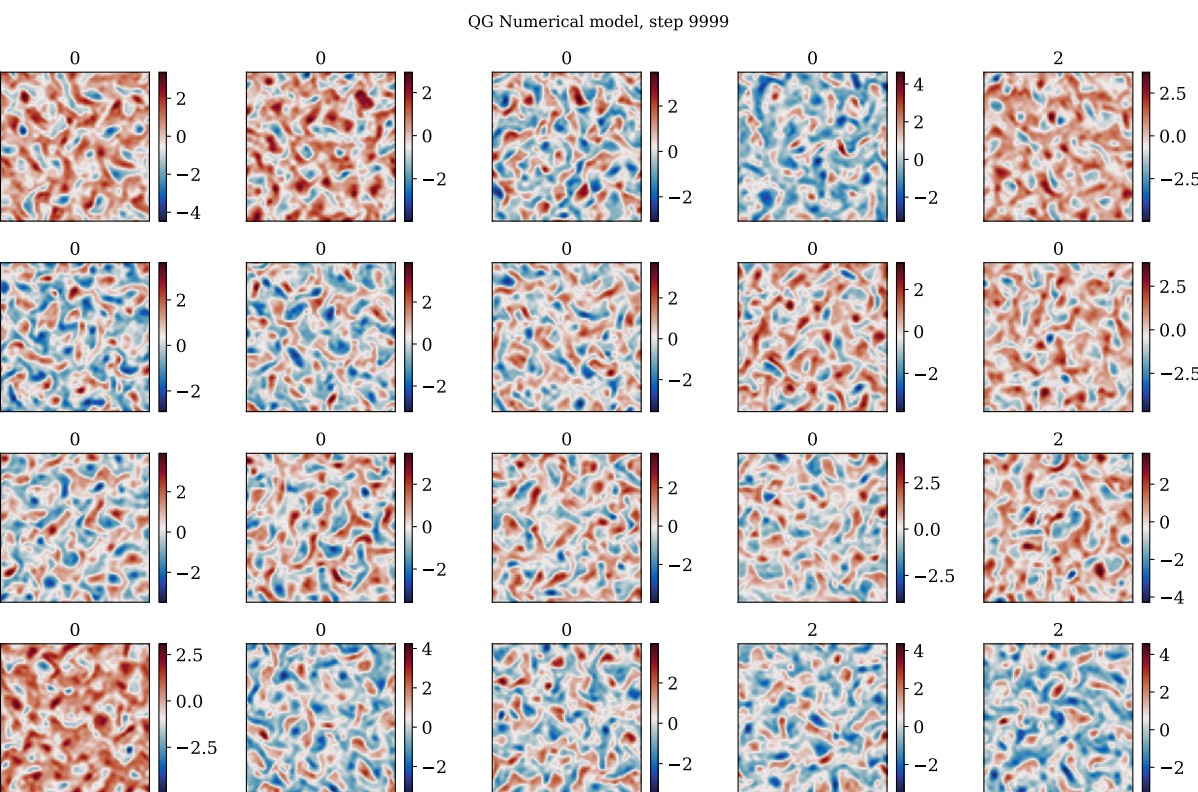

*Figure 19.* QG upper layer potential vorticity snapshots after $10,000$ steps, from the numerical model. The predicted noise level is shown above each fluid state.

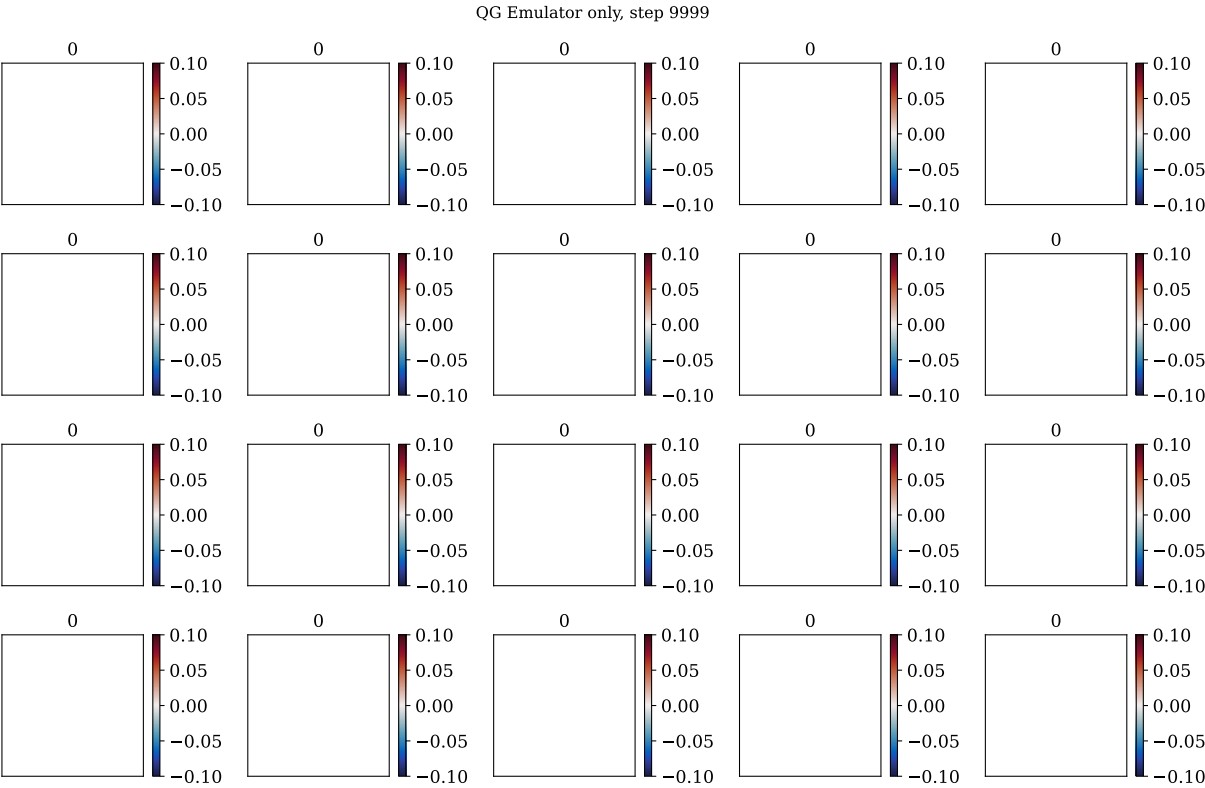

*Figure 20.* QG upper layer potential vorticity snapshots after 10, 000 steps, from the neural network emulator. The predicted noise level is shown above each fluid state.

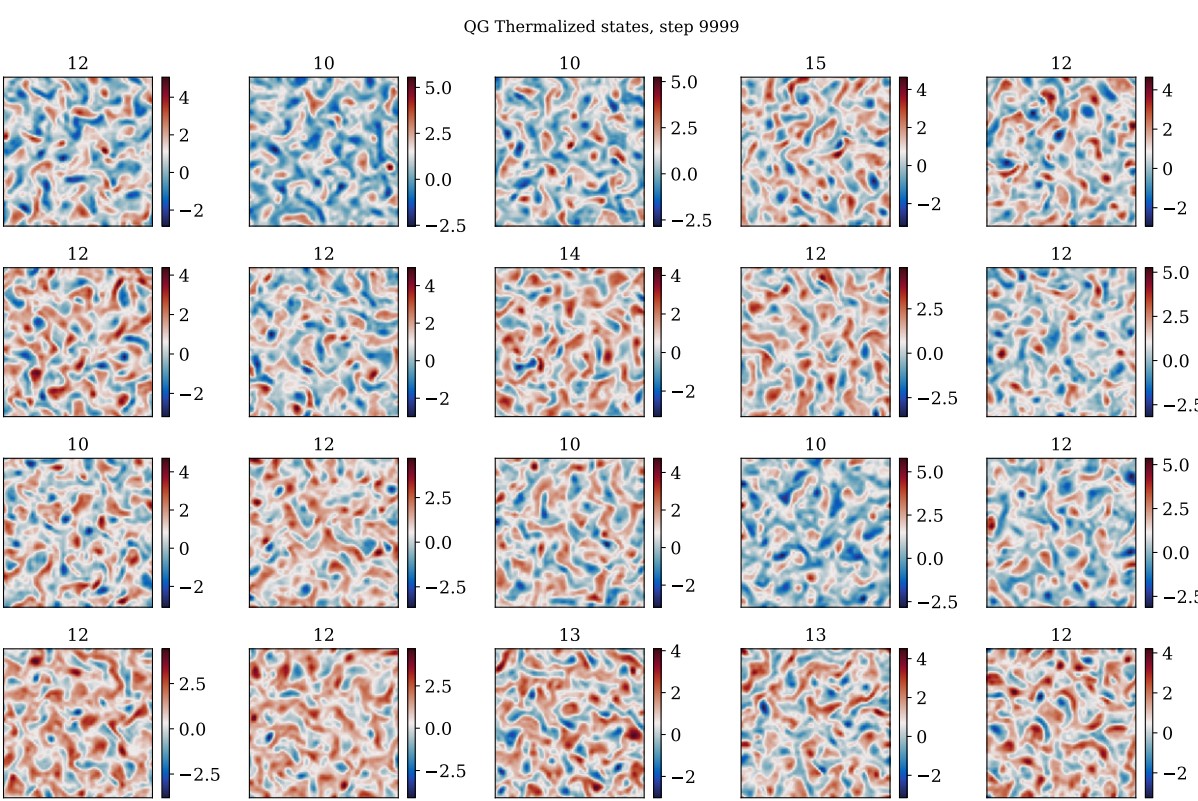

*Figure 21.* QG upper layer potential vorticity snapshots after $10,000$ steps, for the thermalized trajectories. The predicted noise level is shown above each fluid state.

