# OpenReview forum: "Thermalizer: Stable autoregressive neural emulation of spatiotemporal chaos"
_ICML.cc/2025/Conference — ICML 2025 poster_

### Official Review · Reviewer_vKpF · 2025-02-24

**Overall Recommendation:** 4

**Summary:**

This paper introduces "thermalization," a novel inference-time stabilization method for autoregressive emulators of chaotic spatiotemporal systems. It leverages diffusion models, trained separately on the system's invariant measure, to denoise the emulator rollouts during inference time, pulling trajectories back to the equilibrium distribution and preventing divergence. Thermalization is modular, requiring separate training of the emulator and diffusion model, offering an advantage over methods modifying emulator training for stability. Experiments on 2D Kolmogorov flow and quasi-geostrophic turbulence demonstrate significantly extended stable prediction horizons (over 100K emulator steps).

## Update after rebuttal

My score remains the same.

**Claims And Evidence:**

Central Claim: Thermalization extends the stable prediction horizon of autoregressive emulators for chaotic systems.
Evidence is validation on experiments on 2D Kolmogorov flow and QG turbulence, which are highly chaotic and complex PDEs.

**Essential References Not Discussed:**

N/A

**Experimental Designs Or Analyses:**

N/A

**Methods And Evaluation Criteria:**

Evaluation metrics are kinetic energy spectra and mean Squared Error (MSE), which are appropriate for Navier Stokes and QG systems.

**Other Comments Or Suggestions:**

N/A

**Other Strengths And Weaknesses:**

Strengths:
* Novel and Effective Method: Thermalization is a conceptually novel and empirically effective approach to stabilize autoregressive emulators.
* Modular Training: The separation of emulator and diffusion model training is a significant strength, simplifying the training process.
* Strong Empirical Validation: The method is rigorously tested on two challenging turbulent flow problems with convincing results.

Weaknesses:
* Dependence on Diffusion Model Quality: The effectiveness of thermalization relies on the quality of the trained diffusion model and its ability to capture the invariant measure. It's not clear whether such a time-stationary invariant measure exists for many practical systems (such as weather and climate), or whether diffusion model can capture the invariant measure. However, it is a promising direction and worth investigating further.

**Questions For Authors:**

- How does the computational cost of training the diffusion model compare to the cost of training the emulator?
- Can the diffusion model be adapted to consider non-stationary dynamics, or slowly varying invariant measure -- such as those in climate systems?

**Relation To Broader Scientific Literature:**

The paper directly addresses the problem of long-term instability in autoregressive neural emulators, which is a well-recognized issue in the field of neural network emulators of PDEs, especially for chaotic systems.

**Theoretical Claims:**

N/A

---

> ### Author Rebuttal · Authors · 2025-04-01
>
> Thank you very much for the feedback and insightful comments.
>
> > “Dependence on Diffusion Model Quality. The effectiveness of thermalization relies on the quality of the trained diffusion model and its ability to capture the invariant measure. It's not clear whether such a time-stationary invariant measure exists for many practical systems (such as weather and climate), or whether diffusion model can capture the invariant measure.
>
> This is absolutely correct - the ability of the diffusion model to learn the invariant measure is the limiting component of this framework.
>
> > How does the computational cost of training the diffusion model compare to the cost of training the emulator?
>
> For the Unet emulator (which is the same architecture as we use to implement the diffusion model), the emulator training is slightly longer; 36 hours vs 24 hours for the diffusion model, on an A100. However, for the DRN emulators, training takes in the region 10 hours due to the much smaller number of parameters.
>
> > Can the diffusion model be adapted to consider non-stationary dynamics, or slowly varying invariant measure -- such as those in climate systems?
>
> Indeed this is the target for future work - conditioning the diffusion model on some time-varying system by some forcing or time-varying parameters, to allow this framework to be applied to non-stationary systems.
>
> - Additionally, at this anonymous link: https://drive.google.com/drive/folders/1b1DHvR-LJvIZtlpvr6hIRQzxn3JKEuwQ?usp=sharing please find two animations of 6 trajectories for Kolmogorov and QG, where we compare the 3 models, and the temporal flow of the thermalizer trajectories is consistent with the numerical model long after the emulator trajectories have gone unstable.In the bottom row of the QG animation, we show the number of thermalization steps over the past 100 steps, during the rollout, such that the adaptive nature of thermalization is shown. A de-anonymised link to animations will be included in the final version of the paper, as the preservation of this temporal flow is a crucial success of our algorithm.

---

### Official Review · Reviewer_ZXw4 · 2025-03-13

**Overall Recommendation:** 2

**Summary:**

The authors proposed using their method, thermalizer,  to make autoregressive emulator rollouts of chaotic systems more stable. This method relies on a diffusion model, stabilising the emulator's overall predictions during the inference phase. UNet and Dilated ResNet were used as emulators. To verify the quality of the proposed algorithm, the authors used two turbulent systems, Kolmogorov flow and Quasi-geostrophic turbulence. As a result, the thermalized method shows the stability of the predictions over 1e5 timesteps.

**Claims And Evidence:**

The authors proposed a Thermalizer algorithm and showed its effectiveness on Kolmogorov flow and Quasi-geostrophic turbulence datasets. The proposed algorithm is based on the Denoising Diffusion Probabilistic Model (DDPM). In particular, authors modified “the standard implementation of the DDPM framework, by adding a classifier output such that the noise level s is predicted by the network, instead of being passed as an input parameter”.

**Essential References Not Discussed:**

Seems like all essential references were discussed in the article.

**Experimental Designs Or Analyses:**

The experiments were carried out correctly and showed the proposed algorithm's effectiveness. Nevertheless, I would additionally recommend trying different emulators.

**Methods And Evaluation Criteria:**

Proposed methods and datasets for evaluation are quite limited.

1. Limited novelty. It seems to me that the main novelty is incremental. As I can see, the authors slightly modified a DDPM algorithm “by adding a classifier output such that the noise
level s is predicted by the network, instead of being passed
as an input parameter.” The overall algorithm idea is strongly based on the DDPM algorithm. Can the authors provide more arguments to justify their novelty?

2. Comparison with previous approaches. Authors propose an algorithm that is “an alternative approach” “to stabilize predictions and improve the modelling of chaotic dynamics (Li et al., 2022; Jiang et al., 2023; Schiff et al.,2024)”. However, there seems to be a lack of comparison between the Thermalizer and those alternatives. I would suggest comparing Thermalizer with two alternative approaches from the literature review.

3. Datasets. The authors used Kolmogorov flow and Quasi-geostrophic turbulence to verify the proposed method as evaluation datasets. Continuing from the previous comment, I recommend comparing Thermalizer on current datasets (Kolmogorov flow and Quasi-geostrophic turbulence) and from the papers mentioned above. For example, authors can use the chaotic Lorenz-63 system and the Kuramoto-Sivashinsky equation from (Li et al., 2022).

4. Real-life applications. It is always interesting to see real-life applications of the proposed method. However, only experiments on the synthetic data are present in the paper. I suggest checking for long-term instability improvement using Thermalizer on climate and weather modelling tasks.

**Other Comments Or Suggestions:**

It seems that authors use the term Thermalizer in several meanings - diffusion model (“To implement the thermalizer as a diffusion model”) and an algorithm to stabilize trajectories (“We introduced the thermalizer, an algorithm for stabilising autoregressive surrogate models leveraging a pretrained diffusion model of the stationary data distribution”). I would suggest to choose only one meaning for the Thermalizer term.

**Other Strengths And Weaknesses:**

Abstract and introduction are carefully written. Pictures are high quality.

**Questions For Authors:**

1. Am I correct that the Thermalizer Algorithm can be interpreted as an extension of the DDPM framework for turbulent systems, with a diffusion model used to predict the noise level?
2. How do you initialize sinit in Algorithm 1?
3. Can you please clarify what the alpha and beta coefficients in Algorithm 1 stand for?
4. The authors compare the thermalized method with the "numerical model". However, the reference for the "numerical model" seems to be absent. Can the authors clarify this issue?

**Relation To Broader Scientific Literature:**

The proposed approach continues DDPM (Ho et al., 2020) and is an alternate to the previous works (Li et al., 2022; Jiang et al., 2023; Schiff et al., 2024).

**Theoretical Claims:**

n/a

---

> ### Author Rebuttal · Authors · 2025-04-01
>
> Thank you very much for taking the time to assess our work, and for the insightful comments:
>
> - “Am I correct that the Thermalizer Algorithm can be interpreted as an extension of the DDPM framework for turbulent systems, with a diffusion model used to predict the noise level?”
> Not really. Thank you for bringing up this important point, that perhaps was not clearly presented. The DDPM framework provides a generative model of an underlying probability distribution $\pi(x)$ from available samples $x_i \sim \pi(x)$. In the context of sequence generation, all existing DDPM applications have considered *conditional* generative models, of the form $\pi(x_{t+\delta} | x_{t})$, in order to restore temporal consistency of the trajectories.
>  The novelty in our work is that we use a DDPM model only for the *invariant measure* of the system, to stabilise autoregressive rollouts in an adaptive and efficient way; ie we model simply $\pi(x_t)$ for $t$ large. This model just happens to be implemented via a modified DDPM algorithm in this formulation, and it is *unconditional*. Crucially, our method considers separate, independent training components for the emulator and the invariant measure, and we view this as a major advantage over more complex alternatives that require jointly training the components. As we explained in the text, the unconditional aspect of our diffusion model is what guarantees long-time stability (in ergodic systems). Additionally, another contribution of our work is the formulation of the problem of autoregressive error accumulation, which is largely missing from the literature despite extensive empirical studies.
>
> - Comparison with prior work, e.g DySLIM. We agree with the reviewer that this would be an interesting addition to our experimental setup. We are however unable to complete these experiments during this rebuttal period, but will make sure to include them in a later iteration. We emphasize though that such techniques will lead to substantially higher training costs, since the invariant measure needs to be re-estimated every time a parameter is updated.
>
>
> - $s_\text{init}$ and $s_\text{stop}$ were found by running a grid search, much like a hyperparameter search. We ran a search in the range [15,5] where $s_\text{init} > s_\text{stop}$, and ran each combination for 10,000 steps. We chose the best performing run in terms of comparison between the thermalized and numerical model kinetic energy spectra, averaged across all 40 trajectories for Kolmogorov, and 20 trajectories for QG. We have expanded the description of this search in the appendix in the revised manuscript.
>
> - The $\alpha$ and $\beta$ coefficients are the noise-scaling coefficients as presented in the DDPM paper (Ho et al. 2020). Thanks for notifying us that this connection is not explicitly made in the text - we have updated the manuscriptt. As described in the appendix, we use a cosine variance scheduler for our implementation, which uniquely defines alpha and beta for our 1000 noise levels.
>
> - We use a different numerical scheme for the Kolmogorov and QG flows, each are described and referenced in appendices A.1 and A.2 . The QG numerical scheme is our own pytorch implementation, and a github link will be included on de-anonymisation, but a complete description of the method is given in Appendix A2.
>
> - We would like to address your comments on the choice of experiments. In terms of systems to study, we considered Lorenz and Kuramoto-Sivashinky, however these chaotic systems are significantly lower dimensional and less complex than the fluid flows we experiment on. So once the framework was demonstrated on Kolmogorov and QGs, we considered the experiments on Lorenz and Kuramoto-Sivashinsky to be redundant.
>
> - In terms of weather and climate models - indeed, a longer term direction is to apply this framework to full-scale weather and climate models. However this presents a significant data and computational challenge, which is beyond the scope of an initial methodological work. Such an effort would need to be motivated by some initial study and presentation of the new algorithm, which we submit here.
>
> - Additionally, at this anonymous link: https://drive.google.com/drive/folders/1b1DHvR-LJvIZtlpvr6hIRQzxn3JKEuwQ?usp=sharing please find two animations of 6 trajectories for Kolmogorov and QG, where we compare the 3 models, and the temporal flow of the thermalizer trajectories is consistent with the numerical model long after the emulator trajectories have gone unstable.In the bottom row of the QG animation, we show the number of thermalization steps over the past 100 steps, during the rollout, such that the adaptive nature of thermalization is shown. A de-anonymised link to animations will be included in the final version of the paper, as the preservation of this temporal flow is a crucial success of our algorithm.

---

### Official Review · Reviewer_7AYf · 2025-03-13

**Overall Recommendation:** 3

**Summary:**

The goal of this paper is to address the problem of unstable long rollouts by autoregressive neural PDE surrogate models (also called emulators). The core idea is to combine an autoregressive emulator model with an independently trained diffusion model. The role of the autoregressive emulator is then to make an initial prediction of the next system state, whereas the diffusion model 'corrects' this initial prediction to push it towards a data manifold that is in-distribution, thus preventing instabilities. This is implemented through the combination of a denoising diffusion model together with a classifier head that predicts the noise level of the initial classifier prediction, so as to arrive at an appropriate amount of denoising steps. The results, obtained using two model architectures and two datasets, demonstrate that the proposed method maintains stability for long rollouts, whereas the baseline variants diverge.

## Update after rebuttal

Updated recommendation during rebuttal phase.

**Claims And Evidence:**

See the below fields.

**Essential References Not Discussed:**

A recent work [7], accepted for publication in ICLR, proposes 'iterative refinement', a method of which the essence is similar to the method proposed in this paper. As far as I can tell, the major differences lie in that [7] uses Tweedie's formula to iteratively denoise the prediction, and that it decides on a denoising schedule using greedy optimization rather than a separate noise level classifier. The work of [7] was developed concurrently, and I do not doubt that the authors independently arrived at similar ideas. Still, it would be beneficial to inform the reader on the differences and similarities in an updated version of the paper.

[7] Shehata et al. (2025). Improved Sampling Of Diffusion Models In Fluid Dynamics With Tweedie's Formula. https://openreview.net/forum?id=0FbzC7B9xI

**Experimental Designs Or Analyses:**

* The considered metrics and statistics consist of traditional ML metrics (like MSE), physics-based metrics (energy spectra), as well as qualitative results that clearly demonstrate the divergence of the baseline models and stability of the thermalized rollouts, which makes an appropriate analysis strategy for the goal of the paper.

* The considered baseline models are relatively straightforward deterministic emulator models. Although these should definitely be included in the experimental analysis, recently diffusion-based neural PDE emulators have gained significant attention. The method should be explicitly compared against such baselines to establish whether there is any benefit of the approach over existing diffusion-based methods. Consider e.g. (a selection of) the methods studied in [1-4].

[1] Lippe et al. (2023). PDE-Refiner: Achieving Accurate Long Rollouts with Neural PDE Solvers.

[2] Kohl et al. (2023). Benchmarking Autoregressive Conditional Diffusion Models for Turbulent Flow Simulation.

[3] Shysheya et al. (2024). On conditional diffusion models for PDE simulations.

[4] Cachay et al. (2023). DYffusion: A Dynamics-informed Diffusion Model for Spatiotemporal Forecasting.

**Methods And Evaluation Criteria:**

The method is appropriate for the problem at hand, and has the advantage that the autoregressive emulator can be used in combination with an independently (pre-)trained diffusion model.

The benchmark datasets are suitable: the authors demonstrate that baseline models on this dataset suffer from the rollout instability problem, motivating the approach for these scenarios.

**Other Comments Or Suggestions:**

Minor comments:

* line 267: "Gradients are backpropagated
through the full L timesteps, as done in (Brandstetter et al.,
2022; List et al., 2024)." -- If I'm not mistaken, the method of Brandstetter et al. (2022) only backpropagates the gradients by a single step after a rollout during the training process.

**Other Strengths And Weaknesses:**

Strengths:

* The results convincingly demonstrate the method's effectiveness over baseline emulator models in preserving long rollout stability.
* The paper is clearly written and include extensive background and related work sections explaining the problem statement and diffusion models.

Weaknesses:

* My main concern lies in the lack of comparison against diffusion-based baselines (see experimental design). Although the results convincingly demonstrate that Thermalizer improves long rollout stability over vanilla autoregressive emulators, the empirical results do not demonstrate whether it addresses this issue more effectively than established diffusion-based methods.

* It seems that the long rollout stability comes at the cost of slightly worse short-term forecasting performance.

**Questions For Authors:**

* Can you please read [7], and explain the similarities/discrepancies/advantages/drawbacks of your method compared to iterative refinement?

* Equation 7: In the first line of the MSE objective, should $D_\phi^{(1)}$ not be trained against $\epsilon_i$ as opposed to $s_i \epsilon_i$?

**Relation To Broader Scientific Literature:**

The paper relates to general research on autoregressive neural PDE emulator models. Although these models have shown promise for short-term forecasts, they can become unstable over long rollouts. Early ideas to alleviate this issue included augmenting the input data with noise [5] or performing training over longer horizons [6]. More recently, diffusion-based approaches have shown promise in mitigating this issue as well, e.g. [1-4]. This paper is most closely related to such methods, but relies on independently trained autoregressive and diffusion components, and the diffusion model is applied for an adaptive number of steps at inference time based on a separate classifier head that predicts the noise level of the emulator's initial prediction.

[5] Stachenfeld et al. (2022). Learned Coarse Models for Efficient Turbulence Simulation.

[6] Brandstetter et al. (2022). Message Passing Neural PDE Solvers.

**Theoretical Claims:**

N/A

---

> ### Author Rebuttal · Authors · 2025-04-01
>
> We thank the reviewer for their insightful comments.
>
> - Thank you very much for bringing [7] to our attention, that we indeed missed. It is a very interesting application of diffusion models for fluid dynamics, and it has components related to our method. That said, we want to point out what we believe are fundamental differences between the methodology and the problem setup. In essence, (i) [7] considers parallel sequence generation, where several future frames are generated given the current state, and (ii) it is based on conditional diffusion models, which are inherently exposed to distribution shift as one applies them in an auto-regressive fashion to produce long rollouts (longer than those used for training). Needless to say these will be included in the related work section of the updated version.
>
> - The setup of [7] is concerned with diffusion models for general sequence generation, and introduces two novel schemes to improve the numerical efficiency by reducing the number of function evaluations. In that respect, their main motivation is to speed up existing methods based on conditional diffusion. The main setup they consider is not the single-step autoregressive setting, but rather the parallel sampling setting, where several future states are sampled, conditionally on the current state (fig 6).
>
> - As far as we can tell, all the models considered are conditional, similarly as the PDErefiner, where one conditions on the last available state. As such, they are exposed to distribution shifts as the horizon of the rollout at inference time increases. While the numerical results reported by the authors are indeed impressive, we note that the temporal horizon considered is roughly an order of magnitude smaller than our setting (compare our Figure 3, where we report MSE, with their figure 8, where they report correlation). As emphasized in our text, our main insight is precisely to use an unconditional diffusion model to tame such distribution shifts, by exploiting the ergodicity of the dynamical system. That said, this paper introduces interesting insights (e.g. directly using the tweedie /miyazawa estimator rather than reversing the full diffusion path) which could be interesting to combine in our setting.
>
> - Thanks for pointing out the typo in eq 7 – you are correct!
>
> - Comparison with diffusion-based baselines: While we agree with the reviewer that it would be an interesting addition to the experimental evaluation, from our previous discussion we believe that our setup makes this comparison less critical: while these diffusion-based predictions can indeed improve the stability of point-estimate emulators, they are ultimately going to suffer from long-time instabilities as they drift out of distribution. Moreover, even if they were to become stable to arbitrarily long rollouts (as in our setting), they would incur substantially higher training and inference costs. Finally, as mentioned to reviewer GCvA, we ran experiments with the PDErefiner code, but found them to be as unstable as the regular emulator in our testing conditions, so we decided not to report them. It may be that we did not find the correct hyper-parameter settings, so we decided not to report them. but we note that we are not the only ones who had difficulties setting up PDErefiner (see eg https://openreview.net/forum?id=0FbzC7B9xI section E.1, and https://arxiv.org/abs/2309.01745 Figure 8 and section 4.2).
>
> - Additionally, at this anonymous link: https://drive.google.com/drive/folders/1b1DHvR-LJvIZtlpvr6hIRQzxn3JKEuwQ?usp=sharing please find two animations of 6 trajectories for Kolmogorov and QG, where we compare the 3 models, and the temporal flow of the thermalizer trajectories is consistent with the numerical model long after the emulator trajectories have gone unstable.In the bottom row of the QG animation, we show the number of thermalization steps over the past 100 steps, during the rollout, such that the adaptive nature of thermalization is shown. A de-anonymised link to animations will be included in the final version of the paper, as the preservation of this temporal flow is a crucial success of our algorithm.

---

> > ### Comment · Reviewer_7AYf · 2025-04-02
> >
> > I would like to thank the authors for their clear rebuttal!
> >
> > - regarding reference [7], thank you for the detailed explanation of the differences with your paper. I agree that there is a substantial difference between the proposed methods (and even if this weren't the case, the two works should be considered concurrent).
> >
> > - regarding the comparison to diffusion-based baselines: I understand your conceptual argument that conditional diffusion based models ( $p(x^t | x^{t-1})$ ) will ultimately suffer from the distribution shift problem, and I do not disagree. Still, I think your paper would be a lot stronger if it explicitly showed the benefits of Thermalizer over such methods.
> >
> > Overall, after some reflection and taking the other reviewers' comments and author responses into account, I would not be opposed to the publication of this paper since the method and results are valid, even if it seems like a missed opportunity to not show the benefits over autoregressive conditional diffusion emulators. I will update my score accordingly.

---

### Official Review · Reviewer_GCvA · 2025-03-15

**Overall Recommendation:** 4

**Summary:**

The paper proposes a method to stabilize predictions of an autoregressive surrogate model over long-term rollouts. for that, it proposes to learn the invariant measure with a diffusion model and perform denoising steps at inference with a noise level that is guessed by a classifier. The paper claims that arbitrarily long rollouts can be achieved with their method.

**Claims And Evidence:**

The main claim of the paper is well supported by experiments, but should be better precised and tested (see suggestions below).

**Essential References Not Discussed:**

None

**Experimental Designs Or Analyses:**

Visualizing the rollout trajectories and the kinetic energy spectrum over time makes sense for analyzing the divergence of a predicted trajectory (figure 1, 2, 5). Tracking the mse between the stabilized and numerical model is important, as well as the number of thermalization steps (figure 3, 4). i think figure 3 should be complemented with an additional important measure (see suggestions below).

**Methods And Evaluation Criteria:**

The datasets chosen are challenging enough, and the experiments conducted to assess the stability of the rollouts are convincing but should be extended (see suggestions below).

**Other Comments Or Suggestions:**

- Figure 3: the mse converges to a flat level. I believe this is the average mse between an independently generated state from the diffusion model and the observed trajectory. i think you should plot such a constant level on each graph.
- Figure 3: I would suggest plotting histograms of the relative proportion of thermalization steps vs. emulator steps rather than thermalization steps only, which would be more compelling to understand how much "thermalization" is involved.
- l097: "arbtitrarily"

**Other Strengths And Weaknesses:**

Strengths

- The paper is well written, making it simple to understand the method.
- The formalism of the problem (e.g., optimal transport) sheds an interesting perspective on the problem that can inspire future work on this crucial topic.
- The experiments are compelling.

Weakness

- It seems to me that the paper implicitly relies on the following assumption: the trajectories must be available at a high sampling rate over time. if this were not the case, then the "emulator" would have a harder time predicting next steps and thus make much larger errors at each step, which the "thermalizer" may struggle to correct. It could still project back to the invariant measure but would not preserve the temporal dynamics. The paper would have benefited from exploring an example with trajectories that are subsampled in time.

**Questions For Authors:**

- related to the first comment above, you claim that your method can achieve "arbitrarily long rollouts". However, one very simple way to achieve this is to just generate as many independent realizations with your diffusion model as you want in the future. This gives trajectories that have a negligible probability being an actual trajectory. Even though there is technically no "rollout" in this naive "solution" to the problem, I see no guarantees in your method that the "arbitrarily" long sequence you generate is actually close to a true trajectory (in the sense that it has the correct time dynamics). Could you comment on that? In particular, I think the claim of your paper should be changed since it may lead the reader to think you solve the (very) hard problem of obtaining arbitrarily long realistic trajectories.
- another difference with the work [1] you mention is in how the refinement is done. [1] adds noise to the diverged sample and denoises it back (following the standard generation process with diffusion models), while your method does a direct denoising from a diverged sample.
I guess in your terms, this means the "transport" back to the invariant measure is quite different. Could you comment on that?
Could you actually also show results with [1] on your dataset and models? I would particularly be interested to see how one does on figure 4.

[1] Pde-refiner: Achieving accurate long rollouts with neural pde solvers. Lippe et al. 2023

**Relation To Broader Scientific Literature:**

This paper tackles an important problem, which is the stabilization of rollouts of an autoregressive model for evolving trajectories of a spatiotemporal system. Compared to [1], it possesses the key advantage of being constructed separately from the predictive model that we are trying to stabilize.

[1] Pde-refiner: Achieving accurate long rollouts with neural pde solvers. Lippe et al. 2023

**Theoretical Claims:**

The paper makes no theoretical claims, in the sense that the authors do not prove a new theorem.

---

> ### Author Rebuttal · Authors · 2025-04-01
>
> We thank the reviewer for their insightful comments.
>
> - This is a great point which we discussed internally - indeed a diffusion model can produce realistic samples of the flow fields, so it's important to verify temporal consistency. In Figure 4 we show the autocorrelation over time for all different models, and demonstrate that the correlation between thermalized snapshots of different temporal separations is consistent with that of both the numerical model and the emulator. If one were just generating random snapshots, we would expect an autocorrelation of 0. And indeed if we were “over-thermalizing”, by introducing significant noise into the system, this autocorrelation would decrease faster than the emulator and numerical models. However we see from this figure that temporal consistency is preserved, as the number of corrective steps is miminised by the noise-classifying component of the algorithm. This ensures that only states out of distribution are denoised, and are denoised at the minimal amount to return to the invariant measure.
>
> - Additionally, at this anonymous link: https://drive.google.com/drive/folders/1b1DHvR-LJvIZtlpvr6hIRQzxn3JKEuwQ?usp=sharing please find two animations of 6 trajectories for Kolmogorov and QG, where we compare the 3 models, and the temporal flow of the thermalizer trajectories is consistent with the numerical model long after the emulator trajectories have gone unstable.In the bottom row of the QG animation, we show the number of thermalization steps over the past 100 steps, during the rollout, such that the adaptive nature of thermalization is shown. A de-anonymised link to animations will be included in the final version of the paper, as the preservation of this temporal flow is a crucial success of our algorithm.
>
> - The implementation of the denoising process is similar - indeed we also add noise before performing the denoising (see Algorithm 1) - we experimented without this component and found better performance when this forward process is included. The main difference in our work with respect to the PDE-refiner, is that their denoising is still conditioned on the previous (clean) timestep, and therefore their model is still exposed to accumulation of error and distribution shift as states wander out of distribution. We ran experiments with the PDErefiner code, but found them to be as unstable as the regular emulator in our testing conditions, so we decided not to report them. It may be that we did not find the correct hyper-parameter settings, so we decided not to report them. but we note that we are not the only ones who had difficulties setting up PDErefiner (see eg https://openreview.net/forum?id=0FbzC7B9xI section E.1, and https://arxiv.org/abs/2309.01745 Figure 8 and section 4.2). We are happy to include this remark in the updated text.
>
> - With regard to your comment on the sampling rate - this is an interesting suggestion. We do not study this degree of freedom explicitly, however in Figure 4, we see that the decorrelation time is significantly faster between QG and Kolmogorov, indicating that our emulator step size for QG is significantly larger than for Kolmogorov, with respect to the temporal dynamics of the systems. Yet in both cases, the thermalizer is able to stabilize the flow on multiple emulators. Your suggestion is nonetheless very interesting, and we will certainly explore it.
>
> - Thank you for both of your suggestions on improving the clarity of the figure 3 - we agree with these modifications and will incorporate these into the revised manuscript. We have also included an additional figure explicitly showing the number of thermalization steps over time during a rollout (similar to the bottom row in the QG animation linked above), such that the amount of thermalization is more clearly shown.

---

> > ### Comment · Reviewer_GCvA · 2025-04-04
> >
> > I thank the authors for their clear rebuttal. I will raise my score accordingly.

---

### Decision · Program_Chairs · 2025-05-01

**Decision:**

Accept (poster)

**Comment:**

This work addresses the issues wherein running surrogate models for long time-frames might cause degenerate behavior, requiring corrective measures. The authors propose a diffusion-model based model and technique, termed "themalization" that prevents the deterioration of the surrogate-model simulation by acting as a "prior" for the autoregressive model. The method is described and applied to two high dimensional dynamical systems. The reviewers generally appreciated the work as novel and performing well on the demonstrated examples. There were a few concerns, including a few missing references and questions about the short-term unrolling accuracy, however the authors responded and I believe that the paper is correct and significant enough for inclusion in ICML.